# SPRO: Improving Image Generation via Self-Play

**Ritika Jha**⋆ [A] [IIITD]    **Aanisha Bhattacharyya**⋆ [UB] [IIITD]    **Yaman K Singla**⋆ [A]

**Rajiv Ratn Shah** [IIITD]    **Changyou Chen** [UB]    **Balaji Krishnamurthy** [A]

[A]Adobe Media and Data Science Research, [IIITD]IIITD, [UB]SUNY at Buffalo

✉ behavior-in-the-wild@googlegroups.com

## Abstract

Recent advances in diffusion models have dramatically improved image fidelity and diversity. However, aligning these models with nuanced human preferences -such as aesthetics, engagement, and subjective appeal remains a key challenge due to the scarcity of large-scale human annotations. Collecting such data is both expensive and limited in diversity. To address this, we leverage the reasoning capabilities of vision-language models (VLMs) and propose Self-Play Reward Optimization (SPRO), a scalable, annotation-free training framework based on multimodal self-play. SPRO learns to jointly align prompt and image generation with human preferences by iteratively generating, evaluating, and learning to refine outputs using synthetic reward signals such as aesthetics and human engagement. This self-improving feedback loop eliminates the need for external supervision. SPRO comprises three stages: (1) SPRO-Prompt, which trains a Guider-VLM via self-play to generate diverse, high-reward prompts targeting objectives such as PickScore (user preference), LAION-Aesthetics, and EngageNet (engagement); (2) SPRO-Image, which fine-tunes the diffusion model on high-reward images derived from these prompts; and (3) SPRO-Multimodal (SPRO-MM), which integrates both components for full end-to-end alignment. Without relying on human-labeled data, SPRO achieves an average 30% improvement across preference objectives. Moreover, its generated prompts generalize across both open- and closed-source diffusion models. Through iterative self-play, SPRO discovers prompting strategies rarely authored by humans such as emphasizing visual harmony for aesthetics or leveraging shadow-based cues for engagement. SPRO offers a scalable path toward aligning generative models with complex subjective human values.

## 1 Introduction

Recent advances in diffusion models have transformed image generation, enabling the creation of highly realistic and diverse visuals [7]. State-of-the-art systems such as DALL·E [20], Stable Diffusion [21], and SDXL [17] have demonstrated remarkable capabilities across tasks including text-to-image synthesis, style transfer, and image inpainting. Despite these advances, diffusion models remain fundamentally limited in their ability to align with nuanced human preferences—such as aesthetic appeal, engagement, and subjective taste. These dimensions of alignment are inherently abstract, context-dependent, and difficult to quantify, making them challenging to capture through conventional supervised training paradigms.

A number of approaches have been explored to enhance alignment between generated images and human preferences, broadly falling into two categories: *prompt optimization* and *model optimization*.

---

⋆Equal Contribution. Contact behavior-in-the-wild@googlegroups.com for questions and suggestions.

39th Conference on Neural Information Processing Systems (NeurIPS 2025).

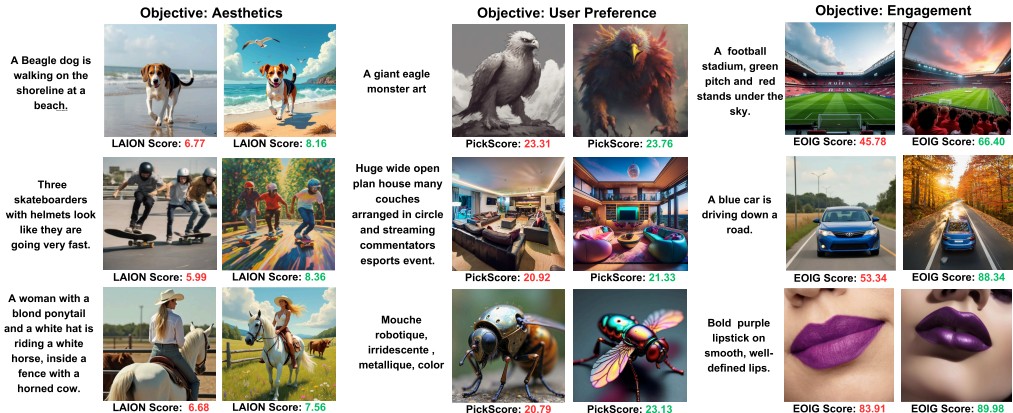

Figure 1: Visual comparison between images generated from original captions (left) and those optimized using our SPRO (Self-Play Reward Optimization) framework (right). SPRO generates images that are more aligned with diverse human preferences - aesthetic appeal, user preference, and engagement, without relying on any human-annotated data. The effectiveness of SPRO is measured using specialized scorers for each objective LAION Score [22] for aesthetics, PickScore [11] for user-preference, and EngageNet (EOIGScore) [9] for engagement.

Prompt optimization methods steer generation by refining textual inputs while keeping the diffusion model frozen [6, 15, 29]. In contrast, model optimization approaches directly fine-tune diffusion weights using preference-based feedback while holding other components constant [1, 5, 27]. Both directions, however, depend heavily on manually curated datasets containing prompts and their preferred or human-improved variants. This reliance on human-authored data imposes fundamental limitations: designing prompts that consistently yield high-quality outputs is difficult due to the nonlinear and often unintuitive relationship between linguistic phrasing and visual realization. For instance, subtle textual changes, such as adding terms, like "highly detailed," "ultra realistic," or invoking specific artists' styles, can dramatically alter visual quality in unpredictable ways. Consequently, models trained on curated datasets tend to replicate only *human-discoverable strategies*, constraining their creative potential. Moreover, large-scale human curation is costly and yields datasets with limited scope and diversity, further restricting progress toward robust preference alignment.

A promising direction for reducing reliance on human-annotated data is the autonomous generation of high-quality synthetic data that capture diverse dimensions of human preference. One mechanism that enables this is *self-play*, a reinforcement learning paradigm in which a model iteratively generates, evaluates, and learns from its own outputs to improve over time. This paradigm removes the need for explicit human supervision by establishing structured feedback loops through which a model refines its behavior. Originally introduced in TD-Gammon [26], where a neural network learned to play backgammon by competing against itself, self-play later achieved widespread prominence through AlphaGo and AlphaZero [24, 25], which reached expert-level performance in games like Go and Chess without relying on human gameplay. More recently, the concept has been extended to large language models (LLMs) for complex reasoning tasks such as mathematics and programming. For example, DeepSeek-R1 [4] employs a reinforcement learning framework in which the model synthesizes its own examples across multiple iterations, boosting pass@1 accuracy on the AIME 2024 benchmark from 15.6% to 71.0%. Similarly, SPIN [3] demonstrates that self-play can enhance instruction-following by generating tasks, solving them, scoring outputs, and fine-tuning on high-quality completions, yielding over a 10% improvement on benchmarks such as GSM8k and TruthfulQA.

In the image domain, self-play has also been explored through SPIN Diffusion [32], which finetunes diffusion models by having them compete against earlier versions of themselves in an iterative general-sum minimax game. This setup allows the model to progressively improve without requiring human-annotated preference labels. SPIN Diffusion is evaluated on two human preference objectives. For user preference, it achieves a score of 22.00 and for aesthetic appeal, it reaches a score of 6.24, after three iterations, showing clear improvement over the base SDXL model which scores 20.99 and 5.67 respectively.

A common theme across recent advances in both text and image domains is the use of self-play to iteratively refine models through synthetic feedback, thereby reducing reliance on human-annotated data. Despite its success, most existing approaches remain confined to a single modality and lack mechanisms for discovering strategies that generalize across architectures or preference objectives.

In language models, self-play has enabled scalable exploration of reasoning strategies and shown strong gains on tasks such as mathematical problem solving and code generation. However, these applications are largely restricted to well-defined, simulator-friendly tasks with unambiguous reward structures, limiting their broader applicability to open-ended generative domains. In the visual domain, approaches such as SPIN Diffusion [32] demonstrate that diffusion models can learn from their own checkpoints, but remain constrained to image space, exhibit poor cross-model generalization, and require large sample budgets (over 500k images) to achieve modest aesthetic gains. These observations highlight the need for a unified framework that combines the strengths of prompt-space exploration and image-space optimization within a multimodal self-play paradigm that is capable of aligning both language and vision models toward complex, human-centered objectives.

We introduce **Self-Play Reward Optimization (SPRO)**, a unified framework that aligns diffusion models with human preferences through multimodal self-play. At its core, SPRO employs a vision–language model (VLM), referred to as the *Guider-VLM*, to autonomously explore and generate optimized prompts via iterative self-play. These optimized prompts are then used to fine-tune a diffusion model, producing high-quality prompt–image pairs aligned with diverse human preference objectives—all without human-annotated data. By coupling language-based reasoning with visual generation, SPRO enables coordinated improvement across modalities and promotes the discovery of strategies that extend beyond human-authored prompting patterns.

SPRO operates in three complementary stages, **prompt-space**, **image-space**, and **joint-space** optimization. In **SPRO-Prompt**, the Guider-VLM takes an image–caption pair as input and generates diverse, reasoning-augmented prompts. Including explicit reasoning chains improves exploration and prompt diversity (Table 5). Each prompt is passed to a frozen diffusion model, which generates corresponding images that are scored by reward models such as PickScore [11], LAION-Aesthetics [22], or EngageNet [9]. High- and low-reward samples form contrastive datasets used to fine-tune the Guider-VLM via Direct Preference Optimization (DPO) [19], enabling it to iteratively learn more effective reasoning and prompting strategies in a vast search space.

In **SPRO-Image**, the diffusion model is fine-tuned using synthetic, high-reward images generated through SPRO-Prompt and their corresponding base captions. This stage improves preference alignment directly in image space without relying on additional prompt tuning.

Finally, **SPRO-Multimodal (SPRO-MM)** integrates both components: the Guider-VLM continues optimizing prompts while the fine-tuned diffusion model generates images. Unlike SPRO-Prompt (which uses a frozen generator) or SPRO-Image (which fixes prompts), SPRO-MM leverages the full capacity of both models, achieving scalable, annotation-free alignment across modalities.

We evaluate SPRO across three key human preference objectives: aesthetic appeal, engagement, and user preference. On SDXL, SPRO achieves win rates of 99.42% for aesthetic appeal (LAION-Aesthetics), 71.60% for engagement (EngageNet), and 65.70% for user preference (PickScore). Unlike prior methods such as SPIN Diffusion [32], which optimize only the diffusion model, SPRO introduces reasoning-augmented self-play via the Guider-VLM, yielding a 0.42-point gain in PickScore and a 1.97-point improvement in aesthetic appeal while using less data. This demonstrates that agent-driven self-play enables more efficient exploration of prompt and image space.

On average, SPRO outperforms both model- and prompt-based baselines, including CAPO, DDPO, and Promptist (Tables 1, 2, and 3). The trained Guider-VLM also produces diffusion-agnostic prompts that generalize effectively across both open- and closed-source models, achieving win rates of 78.9% on SDXL [17], 80.9% on Flux [2], and 67.67% on DALL·E [16], all without additional tuning. In human evaluations, images generated using SPRO prompts were preferred over those from GPT-4o in 66.7% of cases and over base captions in 78.9% (Table 22), confirming stronger perceptual alignment with human preferences.

We make the following key contributions:

- **Multimodal, Multi-Model Self-Play:** To the best of our knowledge, this is the first work to apply a self-play paradigm across multiple models and modalities for preference alignment. SPRO enables coordinated optimization between a vision–language model and a diffusion model, demonstrating the potential of self-play beyond single-domain applications.
- **State-of-the-Art Alignment Across Three Human Preferences:** Through extensive experiments, we show that SPRO achieves state-of-the-art performance on three human preference objectives:

*aesthetics*, *engagement*, and *user preference*. SPRO surpasses all baselines by an average of 30% across these objectives, without relying on any human-annotated data.

- **Intelligent Exploration via the Guider-VLM:** The Guider-VLM drives intelligent exploration in prompt space, automatically discovering novel and often counterintuitive prompting strategies that improve alignment. This results in broader and more efficient optimization compared to traditional human curated data-driven approaches.
- **Reasoning Improves Generation Quality:** We demonstrate that incorporating reasoning chains during prompt generation yields more optimal outputs than scaling compute alone achieving an average 2% improvement across all preference objectives (Table 5).
- **Diffusion-Model Agnostic Framework:** SPRO operates independently of the underlying diffusion architecture, enabling seamless integration with any pretrained model. It achieves a 73.59% average win rate when applied to other open- and closed-source generators.
- **Large-Scale Synthetic Preference Dataset:** We release a synthetic dataset of over one million image–prompt pairs aligned with human preferences, generated entirely through self-play using SPRO. This dataset provides a scalable, annotation-free resource for future research in preference alignment. The dataset is available here.

## 2 Method

The goal of our Multi-Model, Multi-Modal Self-Play Reward Optimization (SPRO) framework is to optimize images for various human preference objectives, such as aesthetic appeal, engagement, or user preference. Given a user-provided base image, caption and target objective, the framework produces more refined output images while preserving the original intent. Our method operates through three progressive stages, each targeting a distinct axis of optimization: prompt-space optimization, image-space optimization, and joint optimization. Next, we cover the formal definition of the optimization problem and describe SPRO framework.

### 2.1 Problem Formulation

To operationalize our framework, we formalize SPRO as an optimization problem over prompt and image generation spaces. Each stage corresponds to a distinct optimization objective with respect to the model components and the reward function.

Let $\mathcal{D}_\theta$ denote a diffusion model with parameters $\theta$, $p$ a prompt generated by a Guider VLM, and $\mathcal{R}_o$ a reward function encoding human preferences as aesthetic appeal, engagement or user preferences. We define the optimization objectives as follows:

- **SPRO-Prompt:** We optimize prompts $p$ while keeping the diffusion model parameters $\theta$ fixed, allowing the system to discover effective conditioning strategies.

$$\max_p \ \mathbb{E}[\mathcal{R}_o(\mathcal{D}_\theta(p))] \tag{1}$$

- **SPRO-Image:** We fine-tune the diffusion model parameters $\theta$ using high-quality synthetic images generated from optimized prompts $p'$ obtained in the previous stage. During this process, the prompts $p$ are kept fixed (to the original prompts), allowing the model to better align its generations with the target objective $O$ in the image space using self-play data.

$$\max_\theta \ \mathbb{E}[\mathcal{R}_o(\mathcal{D}_\theta(p))] \tag{2}$$

- **SPRO-MM:** We jointly leverage both the optimized prompts $p'$ and the diffusion model weights $\theta$, allowing for co-evolution and deeper alignment:

$$\max_{p',\theta} \ \mathbb{E}[\mathcal{R}_o(\mathcal{D}_\theta(p'))] \tag{3}$$

### 2.2 SPRO-Prompt

Self-play refers to the process of progressively improving a model by learning from the outputs of its own previous iterations. In the **prompt-space optimization** stage, our *Self-Play Reward Optimization (SPRO-Prompt)* framework leverages this paradigm by training a *Guider-VLM* through self-play to generate optimal prompts for a given human preference objective. The Guider-VLM receives an

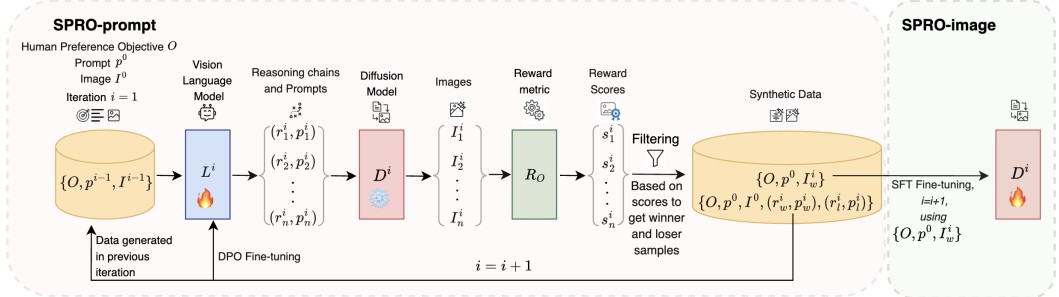

Figure 2: Our SPRO framework consists of three stages: **SPRO-prompt** (Left): It iteratively finetunes a VLM $L^i$ using DPO to generate reasoning chains $r_j^i$ and prompts $p_j^i$ conditioned on human preferences $O$, base prompts $p^0$, and images $I^0$ for 1st iteration and in further iterations it uses prompts and images generated in previous iteration. These prompts guide a diffusion model $D^i$ to produce images $I_j^i$, which are scored by a reward function $R_O$, resulting in scores $s_j^i$. High-reward samples are filtered to form synthetic data. The **SPRO-image** (right) fine-tunes the diffusion model $D^i$ for objective $O$, using base prompts $p^0$, and corresponding winner image $I_w^i$ to directly align image generation with human preferences.

image, its base caption, and a target objective as input, and produces both reasoning chains and optimized prompts as output.

SPRO-Prompt operates in two phases. **(1) Synthetic data generation:** the Guider-VLM produces diverse reasoning chains and prompt candidates, sampled at varying temperatures to encourage exploration. Each prompt is passed to a frozen diffusion model to generate images, which are then evaluated by a trained reward model. Samples with rewards below the mean of the distribution are discarded, and CLIP similarity [18] is used as a sanity check to ensure that generated images remain semantically consistent with the base image (see Table 9). **(2) Direct Preference Optimization (DPO)** [19]: the Guider-VLM is fine-tuned using contrastive feedback derived from the synthetic data, enabling it to iteratively learn more effective reasoning patterns and prompt strategies tailored to each objective. The overall SPRO-Prompt pipeline is illustrated in Figure 2 and is applied across multiple human preference objectives, including aesthetic appeal, engagement, and user preference.

### 2.2.1 Synthetic Data Generation

For a given human preference objective $O$, we begin with a set of image-base caption pairs. In the first iteration (i.e., $i = 1$), let these base caption pairs be denoted by $p^0$ and $I^0$, belonging to the dataset $X^0$.

1. A VLM $L^i$ takes the image-caption pair $(p^0, I^0)$ as input, along with the details of the target human preference $O$ as a prompt, and generates a set of $n$ candidate reasoning chains and corresponding improvised prompts for the diffusion model, denoted as $(r_j^i, p_j^i)$ for $j \in \{0, \ldots, n\}$, sampled at different temperatures to ensure diversity.
2. The diffusion model $D$ now generates images using the candidate prompts:
$$I_j^i = D(p_j^i) \quad \forall j \in \{0, n\} \tag{4}$$
3. Each image is evaluated using a reward function $R_o$ for that objective:
$$s_j^i = R_o(I_j^i) \quad \forall j \in \{0, n\} \tag{5}$$

### 2.2.2 Iterative DPO Refinement

1. For a given objective $O$, in iteration i, we generate a set of prompts $p_j^i$ for $j \in \{0, \ldots, n\}$, from the Guider VLM $L^i$, each of which is used to generate an image via a frozen diffusion model. These images are then scored using a trained reward model, and the corresponding reward scores are used to rank the prompts.
2. In each iteration i, we select the highest- and lowest-reward prompts, $(p_w^i, p_l^i)$, based on their associated image scores, and use them as winner–loser pairs for training. Samples are discarded if the reward improvement is below the mean of the generated distribution. To ensure semantic consistency, we also filter out prompt–image pairs with CLIP similarity below 0.75, ensuring comparisons focus on alignment with the objective while preserving original content. This filtered dataset, after applying the threshold condition, becomes the new dataset $X^i$.

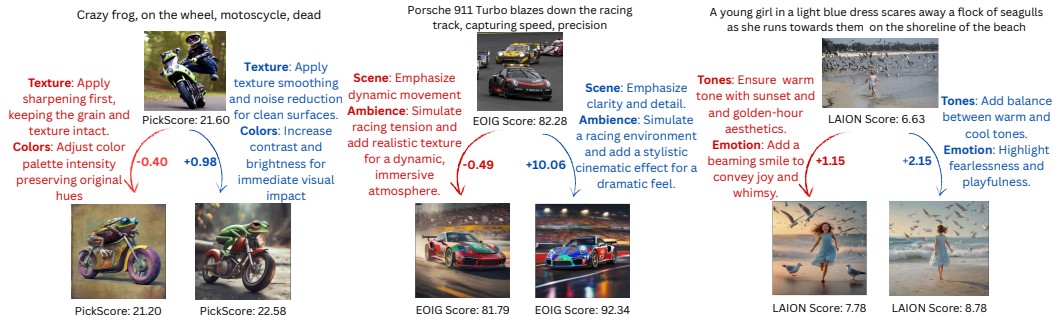

Figure 3: For each base caption and image (top), SPRO-Prompt generates multiple reasoning chains and refines the prompt for the diffusion model. Red arrows indicate lower reward generations; blue arrows denote higher reward generations. SPRO discovers novel reasoning strategies targeting aspects like texture, color, etc- enabling reward-driven image refinement through self-play. More samples of strategies in Table 21

3. We fine-tune the Guider VLM $L^i$ using Direct Preference Optimization on the preference pairs in $X^i$ to obtain the updated model $L^{i+1}$:

$$\mathcal{L}_{\text{DPO}}(L^i; L_{\text{ref}}) = -\mathbb{E}_{(R_l, R_w)} \left[ \log \sigma \left( \beta(R_l - R_w) \right) \right] \quad (6)$$

$$\text{where } R_l = \log \frac{L^i((r_l, p_l)|I, C, O)}{L_{\text{ref}}((r_l, p_l)|I, C, O)}, \quad R_w = \log \frac{L^i((r_w, p_w)|I, C, O)}{L_{\text{ref}}((r_w, p_w)|I, C, O)} \quad (7)$$

$L_{\text{ref}}$ is the reference model (pre-fine-tuning), and $\beta$ is a hyperparameter.

We optimize the Guider VLM iteratively, using the fine-tuned model from iteration $i-1$ to generate preference data for training iteration $i$. This process, forms a positive feedback loop that gradually shifts the model's output distribution toward higher rewards in the preference space. Across different objectives, we observe consistent improvements in reward metrics over successive iterations, with performance typically saturating after 3–4 rounds. This iterative refinement enables the Guider VLM to discover novel prompting strategies aligned with the target objective that are often beyond human intuition and are not constrained by any fixed dataset, producing preferred prompts that yield improved images while preserving user intent.

## 2.3 SPRO-Image

For image-space optimization, we extend the exploration conducted in prompt-space optimization via SPRO-prompt. Given a target objective $O$, we fine-tune the diffusion model using only high-reward (winner) synthetic images generated through the iterative self-play process. This approach directly aligns the diffusion model with human preference objectives, allowing it to produce optimized images even from simple base prompts. As illustrated in Figure 2, our pipeline leverages SPRO-prompts obtained through self-play to drive image-level fine-tuning.

1. We leverage the SPRO-Prompt to get best image $I_w^i$ for entire training data in each iteration.
2. These high-reward images are paired with their corresponding base captions $\{p^0, I_w^i\}$ to form a synthetic training dataset $S_{\text{i train}}$ for each iteration $i$.
3. We fully finetune the diffusion model $\theta$ to minimize the reconstruction loss between images generated by base captions and the reward-optimized targets for every iteration i:

$$\mathcal{L}_{\text{SRPO}}(\theta) = \mathbb{E}_{(p^0, I_w^i) \sim \mathcal{S}_{\text{i train}}} \left[ \|\mathcal{D}_\theta(p^0) - I_w^i\|^2 \right] \quad (8)$$

## 2.4 SPRO-MM

In the joint optimization stage, **SPRO-MM**, we combine the strengths of optimized prompts and a finetuned diffusion model. Unlike **SPRO-Prompt**, where the diffusion model remains frozen and only the Guider VLM is trained to produce optimized prompts, or **SPRO-Image**, where the diffusion model is finetuned using reward-aligned synthetic high quality images generated while keeping prompts fixed, SPRO-MM uses both the Guider VLM to generate optimized prompts for a given objective $O$, and these prompts are then input into the diffusion model that has already been finetuned for the same objective, using self-play data.

## 3 Experiments

We conducted experiments to evaluate the effectiveness of our self-play framework, SPRO, to optimize images for various human preference objectives. Data-generation and training details are provided in Appendix A.4.

**Reward Metrics** We focus on three human preference objectives: aesthetic appeal, user preference, and engagement, each representing a distinct dimension of evaluation. These objectives are quantified using learned reward models. For user preference, we use PickScore [11], a CLIP-based model trained on large-scale preference data. For engagement, we adopt EngageNet [9], a foundation model trained on Twitter data to predict the social media engagement an image is likely to receive. For aesthetic appeal, we employ the LAION aesthetic scorer [22], trained on human-annotated aesthetic ratings. To assess generalization, we also evaluate on unseen reward models, including VILA [8] and the Improved Aesthetic Scorer [23] for aesthetics, as well as ImageReward [28] for user preference.

**Evaluation** We evaluate our proposed framework on distinct test sets tailored to each human preference objective. For aesthetic appeal, we use a held-out set of 514 images from the Flickr30k dataset. We evaluate on the PartiPrompts [31] and Pick-a-pic test split. For engagement, we adopt the same test set from EngagingImageNet dataset to ensure consistency with prior benchmarks. For user preference, we evaluate on the Pick-a-Pic test set, as well as the PartiPrompts dataset to capture diverse user preferences.

## 4 Results

We evaluate SPRO against state-of-the-art methods in two categories: prompt-only approaches, which optimize text prompts without altering the diffusion model, and diffusion-only approaches, which directly tune the image generator using preference rewards or annotated data. Prompt-only baselines include Promptist [6] and PAE [15], both reliant on real datasets and supervision. Diffusion-only baselines include DDPO [1], SPIN Diffusion [32], and EngageNet [9], which is trained on Twitter engagement-labeled images. We also compare with general-purpose LLMs as prompters like GPT-4o and LLAMA-11B using CoT.

We benchmark all three stages of our framework, SPRO-Prompt, SPRO-Image, and SPRO-MM, on aesthetic appeal, user preference, and engagement. For aesthetics (Table 2), SPRO-Prompt achieves a LAION score of 8.60, outperforming Promptist (7.23), PAE (6.27), GPT-o3 (7.67) and GPT-4o (7.57) and showcases a 34% gain over base captions. For user preference (Table 1), SPRO-Prompt attains a PickScore of 22.42, surpassing both prompt-only GPT-4o (22.31), GPT-o3 (19.47) and diffusion-optimized models (SPIN: 22.00, DDPO: 21.59), demonstrating that synthetic self-play in prompt-space can rival data-intensive methods. On engagement (Table 3), SPRO-Prompt scores 83.48, 88.36, and 90.24 across low, medium, and high engagement buckets more than doubling EOIG's score in the low bucket (38.76 to 83.48), showing SPRO's ability to amplify weak content. Further ablations and experiments are provided in Appendix A.2. Qualitative samples are shown in A.6.

**Synthetic data alleviates the need for large-scale annotation.** Unlike prior optimization methods such as DDPO [1], which relies on explicit pairwise preference annotations and achieves a PickScore of 21.59, our method reaches a higher mean score of 22.42. Similarly, Promptist, which depends on manually curated prompts, attains a LAION aesthetic score of 7.23, while our method achieves 8.60. We also find that diffusion models finetuned on self-play data outperform those trained on real data. For example, on the engagement objective, SPRO-Image delivers an average improvement of 42% over EOIG-PFT, which was trained on images sourced from real tweets. This underscores SPRO's scalability and strength in aligning with human preferences using only synthetic data without costly manual annotation.

**Guider VLM leads to more efficient exploration and broader optimization.** SPRO-Prompt, which uses self-play in prompt-space, achieves higher aesthetic (8.02 vs. 6.05) and user preference scores (22.84 vs. 22.31) than SPIN Diffusion [32], while using only 30k training examples compared to SPIN Diffusion's 500k. This underscores that self-play in prompt space is substantially more sample-efficient than self-play in image space. The guider VLM enables richer and targeted exploration in strategies as shown in Figure 3, suggesting that intelligent reasoning, not scale alone, is key to optimizing human preference objectives. Moreover, when this synthetic data is used to finetune a diffusion model directly, it also outperforms self-play in diffusion space.

| Prompt Model | Approach | PickScore |
|---|---|---|
| Base captions | | 20.99 |
| LLAMA 11B cot | Prompt | 21.96 |
| GPT4o | Prompt | 22.31 |
| GPT-o3 | Prompt | 19.47 |
| Qwen-3 | Prompt | 22.10 |
| DDPO[2] | Image | 21.59 |
| SPIN Diffusion[3] | Image | 22.00 |
| SPRO-prompt(Ours) | Prompt | **22.42** |
| SPRO-image(Ours) | Image | 22.09 |
| SPRO-MM(Ours) | Both | **22.42** |
| Base captions | | 21.89 |
| DDPO[2] | Image | 22.27 |
| SPIN Diffusion[3] | Image | 22.31 |
| CAPO[1] | Image | 22.83 |
| SPRO-prompt(Ours) | Prompt | **22.84** |
| SPRO-image(Ours) | Image | 22.55 |
| SPRO-MM(Ours) | Both | 22.77 |

Table 1: The top block of the table reports results on the Pick-A-Pic (PAP) dataset[11] , while the bottom block presents results on the PartiPrompts [31]. The SPRO-prompt achieves a higher PickScore on the test set of the Pick-A-Pic dataset as well as PartiPrompts than existing diffusion tuning baselines such as CAPO [12], DDPO [1] and SPIN Diffusion [32], indicating stronger alignment with user preference.

| Prompt Model | Approach | LAION score |
|---|---|---|
| Base captions | | 6.40 |
| Prompt Auto Edit[5] | Prompt | 6.27 |
| Promptist[4] | Prompt | 7.23 |
| LLAMA 11B | Prompt | 7.38 |
| GPT4o | Prompt | 7.57 |
| GPT-o3 | Prompt | 7.67 |
| Qwen-3 | Prompt | 7.55 |
| SPRO-prompt(Ours) | Prompt | **8.60** |
| SPRO-image(Ours) | Image | **7.48** |
| SPRO-MM(Ours) | Both | **8.42** |
| Base captions | | 5.67 |
| DDPO[2] | Image | 5.77 |
| Spin Diffusion[3] | Image | 6.05 |
| SPRO-prompt(Ours) | Prompt | **8.02** |
| SPRO-image(Ours) | Image | **7.01** |
| SPRO-MM(Ours) | Both | **8.02** |

Table 2: The top block of the table reports results on the Flickr test dataset [30] , while the bottom block presents results on the PartiPrompts [31]. The SPRO-Prompt,SPRO-Image and SPRO-MM outperform all prompt-space baselines, Promptist [6] and PAE [15] on aesthetic alignment measured using LAION score [22] on both datasets.

**Prompt optimization excels by discovering effective strategies.** A comparison across stages of our framework shows that SPRO-Prompt, which performs self-play in prompt space, consistently uncovers novel strategies that improve outcomes. As shown in Figure 3, the Guider-VLM learns these strategies iteratively through reasoning chains. Table 21 highlights how strategies vary by objective: user-preference prompts benefit from technical and systematic phrasing, while aesthetic prompts perform better with artistic phrasing. For engagement, prompts emphasizing motion blur outperform those using motion lines. These insights underscore the value of strategy exploration in prompt space. SPRO-Prompt outperforms SPRO-Image across all objectives, showing that such strategies are harder to teach through diffusion model finetuning alone. SPRO-MM, which combines both, yields further gains, demonstrating the complementary benefits of prompt and image-space optimization.

**SPRO-Prompt generalizes across diffusion models.** Prompts optimized by the guider VLM through self-play are diffusion-agnostic. Though trained using SDXL-base as the frozen diffusion model, these prompts transfer effectively to both open and closed-source models in zero-shot. Applying the same test-set prompts to Flux [2] and DALL·E 3 [16], we observe consistent improvements over base prompts across all objectives. For aesthetic appeal (LAION score [22]), Flux improves from 6.77 to 7.86 and DALL·E 3 from 6.79 to 7.98. On engagement, using Flux, we observe a 21.5% average gain across all buckets. While DALL·E 3 already performs well in low-engagement scenarios, SPRO-Prompt boosts its quality in medium and high buckets. For user preference (PickScore [11]), SPRO-Prompt improves Flux from 21.99 to 22.20. Results in Table 4 confirm that SPRO-Prompt produces broadly transferable, high-quality prompts.

**Zero-shot generalization across unseen reward models.** SPRO demonstrates robust zero-shot transferability across reward signals, showing that it learns general prompting strategies tied to a preference objective rather than overfitting to the specifics of any particular reward model. For aesthetic appeal, although trained with the LAION scorer [22], it also improves performance on unseen metrics, raising scores from 7.38 to 8.60 on LAION, 60.74 to 65.16 on VILA [8], and 6.30 to 7.47 on the Improved Aesthetic Scorer. For user preference, trained with PickScore [11], SPRO-Prompt not only improves PickScore itself (20.99 to 22.42) but also achieves higher scores on ImageReward [28] (0.65 to 1.02) and CLIP similarity [18] (87.76 to 89.32). Results in Tables 13 and 14 confirm that SPRO-Prompt produces generalizable, high-quality prompts aligned with human preferences. Beyond trained reward models, we also evaluate SPRO in a setting guided only by human aesthetic ratings from the AVA dataset, where SPRO-Prompt improves average scores from 5.90 to 6.80. This experiment, detailed in Appendix A.2.8, highlights SPRO's adaptability even when no external reward function is available.

**Generalization across multiple datasets.** SPRO demonstrates strong robustness when evaluated beyond the training distribution, consistently improving performance across diverse datasets and reward models. For the aesthetic appeal objective, as shown in Table 18, SPRO-Prompt significantly improves scores on the PartiPrompts dataset across three independent evaluators: LAION Score (6.16 to 8.01), VILA (55.15 to 64.81), and the Improved Aesthetic Scorer (5.60 to 6.78). Similarly, for the user preference objective (Table 19), SPRO-Prompt achieves a notable gain in PickScore (21.89 to 22.84), reinforcing its ability to generalize preference-aligned strategies to new distributions of prompts and images. We further validate this finding on the LexicaDB dataset [13], following the RATTPO [10] setup. Since the original test set is unavailable, we sample 160 random instances and optimize them with our user-preference-tuned SPRO-Prompt. As shown in Table 20, SPRO-Prompt achieves an ImageReward score of 1.439, substantially higher than both the base prompt (0.1588) and the RATTPO baseline (1.132). These results demonstrate that SPRO generalizes effectively across datasets and reward models, reinforcing its robustness and applicability to diverse prompt-image distributions.

**SPRO supports multi-objective optimization.** Beyond single-objective training, SPRO can accommodate multiple reward signals, enabling alignment with complementary, orthogonal, or even conflicting preferences. In the first setup, we optimize for both aesthetics (LAION Score) and semantic fidelity (CLIP similarity). As shown in Table 15, SPRO-Prompt improves aesthetics from 6.40 to 7.90 while maintaining stable CLIP similarity (83.19 to 83.28), showing that it can enhance aesthetic quality without compromising semantic consistency. We then consider a more challenging case with orthogonal objectives: aesthetics (LAION Score) and user preference (PickScore), these two metrics represent distinct and potentially unaligned preferences. Table 16 shows that SPRO-Prompt improves both, raising LAION from 6.40 to 7.66 and PickScore from 22.69 to 22.96, demonstrating its ability to balance distinct and potentially unaligned rewards. To further test flexibility, we evaluate SPRO under conflicting objectives by designing a composite pseudo-reward function defined as $2 \times \text{LAION Score} - 0.5 \times \text{PickScore}$, which favors higher aesthetics while penalizing user preference. This setup simulates real-world scenarios where artistic choices may enhance visual appeal but reduce broader audience preference. Using this objective, SPRO-Prompt behaves as intended (Table 17), increasing LAION from 6.40 to 7.62 while reducing PickScore from 22.69 to 21.70. Our findings demonstrate that the SPRO framework effectively supports multi-reward optimization, including both orthogonal and conflicting objective settings.

**SPRO is model-agnostic.** SPRO consistently enhances performance across diverse VLM backbones, demonstrating its robustness independent of model choice. As shown in Table 12, reasoning-based backbones such as Qwen-3-7B achieve the largest improvements, rising from 7.55 to 8.46 in a single iteration. Non-reasoning models also benefit, with LLaVA-7B improving from 6.76 to 7.22 and LLaMA-3.2-11B improving from 7.38 to 8.30. These results demonstrate that SPRO generalizes across reasoning-augmented and standard VLMs, with stronger backbones converging more efficiently but all models consistently gaining from self-play.

**Human Study** To assess alignment with human preferences beyond automatic rewards, we conducted a human evaluation comparing images generated from three prompts: base caption, GPT-4o-optimized prompt, and our SPRO-prompt. For each input, three images were generated using the same diffusion model (SDXL or Flux), and annotators selected their preferred image in a randomized, blind setting. As shown in Table 22, SPRO-prompt was preferred over GPT-4o in 66.7% of cases and over base captions in 78.9%. The human study protocol is detailed in Appendix A.5.

# 5 Related Works

Recent research on aligning text-to-image diffusion models with human preferences falls broadly into two complementary streams: *model-space optimization* and *prompt-space optimization*.

**Diffusion Model Tuning.** Early efforts directly fine-tune diffusion parameters using preference feedback. For instance, DPO-Diff [27] introduces text-guided gradients for efficient optimization within a restricted prompt domain, while Diffusion-DPO [27] removes the explicit reward model and instead optimizes diffusion parameters via implicit rewards. DDPO [1] and DPOK [5] extend this line of work by training reward models on human preference datasets and fine-tuning diffusion models through reinforcement learning. CaPO [12] further calibrates multi-objective rewards to balance competing alignment goals. Although these methods achieve improved alignment, each

| Prompt Model | Approach | EOIG Score | | |
|---|---|---|---|---|
| | | Low | Mid | High |
| Base captions | | 38.76 | 51.69 | 56.14 |
| LLAMA 11B | Prompt | 83.19 | 87.72 | 89.32 |
| GPT4o | Prompt | 77.70 | 87.83 | 89.95 |
| GPT-o3 | Prompt | 75.91 | 75.97 | 79.51 |
| Qwen-3 | Prompt | 74.12 | 83.12 | 84.23 |
| EOIG-PFT[6] | Image | 43.12 | 56.10 | 62.30 |
| EOIG-RLHF-ES[6] | Image | 40.20 | 53.10 | 58.27 |
| EOIG-RLHF-DSG[6] | Image | 39.49 | 52.97 | 57.91 |
| SPRO-prompt(Ours) | Prompt | **83.48** | **88.36** | **90.24** |
| SPRO-image(Ours) | Image | 70.70 | 77.94 | 77.43 |
| SPRO-MM(Ours) | Joint | 83.20 | 87.39 | 89.38 |

Table 3: SPRO-prompt has the highest EoIG Scores [9] which is human engagement on a scale of 0-100. We show results for the three buckets of data samples low, medium and high made based on the EoIG Scores of base images.

| Prompt Model | Diffusion Model | Pick Score | LAION Score | EOIG Score | | |
|---|---|---|---|---|---|---|
| | | | | Low | Mid | High |
| Base captions | SDXL | 20.99 | 6.40 | 64.34 | 71.75 | 75.10 |
| SPRO-prompt(ours) | SDXL | **22.42** | **8.60** | **83.48** | **88.36** | **90.24** |
| Base captions | Flux | 21.99 | 6.77 | 64.53 | 71.80 | 70.80 |
| SPRO-prompt(ours) | Flux | **22.20** | **7.86** | **77.25** | **85.50** | **89.01** |
| Base captions | DALLE-3 | 22.07 | 6.79 | 66.13 | 72.36 | 60.01 |
| SPRO-prompt(ours) | DALLE-3 | 22.07 | **7.98** | 61.68 | **74.69** | **77.98** |

Table 4: Prompts optimized using SPRO-prompt generalize well to other diffusion models as Flux [2] and DALLE-3 [16], in zero-shot, across all three objectives, highlighting the diffusion agnostic nature of our framework

tuned model remains tied to a specific backbone and objective, and their dependence on large-scale human feedback datasets limits scalability.

**Prompt-Space Optimization.** An alternative direction focuses on modifying text prompts while keeping diffusion weights frozen. PromptCoT [29] introduces a Chain-of-Thought (CoT) approach that aligns prompts with high-quality image descriptions and fine-tunes a large language model (LLM) on curated text data. PROMPTIST [6] adapts user-written prompts into model-preferred versions through supervised fine-tuning on engineered examples, followed by reinforcement learning to enhance visual appeal while preserving intent. Dynamic Prompt Optimization [15] employs online reinforcement learning to dynamically re-weight or insert tokens during generation. Despite their effectiveness, these systems rely on human-authored prompt–image pairs and therefore explore only *human-discoverable* strategies.

Across these two lines of research, prompt-based methods overlook image-space fine-tuning, while diffusion-based methods neglect textual conditioning. Our proposed Self-Play Reward Optimization (SPRO) framework unifies both directions: a Guider-VLM autonomously discovers novel—and often counterintuitive—prompting strategies that yield high-reward images, which are then used to further fine-tune diffusion models. In doing so, SPRO bridges the gap between prompt-only and diffusion-only approaches, introducing a scalable, multimodal framework for preference alignment.

# 6 Conclusion

We introduce *Self-Play Reward Optimization (SPRO)*, a unified framework for objective-driven prompt optimization for image generation. SPRO leverages a Guider-VLM to engage in iterative self-play, generating and evaluating diverse reasoning chains and prompts conditioned on base image-caption pairs. By combining prompt and image space optimization in a multi-stage process, SPRO significantly improves alignment with multiple human preference objectives including aesthetics, user preference, and engagements. Unlike prior approaches that rely on human-annotated datasets or handcrafted prompts, SPRO autonomously explores and discovers effective strategies without additional human supervision. Our experiments demonstrate consistent gains in each reward metric, validating SPRO's capability to generalize across objectives.

**Limitations** While promising, SPRO has several limitations. SPRO's effectiveness is bounded by the reliability of external reward models, which act as proxies for human preference but may not fully capture nuanced human preference. Moreover, these human preferences are addressed by SPRO independently, it does not yet support dynamic trade-off optimization in scenarios involving multiple goals (e.g., maximizing aesthetic quality while maintaining high engagement). Future work could explore reward balancing and preference-aware optimization to address these challenges.

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

## A Technical Appendices and Supplementary Material

### A.1 Acknowledgments and Disclosure of Funding

Rajiv Ratn Shah is partly supported by the Infosys Center for AI, the Center of Design and New Media, and the Center of Excellence in Healthcare at IIIT Delhi. This work is partially supported by NSF AI Institute-2229873, NSF RI-2223292, an Amazon research award, and an Adobe gift fund. Any opinions, findings and conclusions or recommendations expressed in this material are those of the author(s) and do not necessarily reflect the views of the National Science Foundation, the Institute of Education Sciences, or the U.S. Department of Education.

### A.2 Ablation

#### A.2.1 Effect of Reasoning-Based Prompting

We conduct an ablation study to evaluate the impact of reasoning-based prompts versus standard prompts across two VLMs. The results (Table 5) show consistent improvements in aesthetics, human preference, and engagement scores when reasoning is incorporated.

| Model | Reasoning | Aesthetics Laion Score | Human Preference PickScore | Engagement (EOIG Score) *Low* | *Mid* | *High* |
|---|---|---|---|---|---|---|
| LLaMA 11B | ✗ | 8.19 | 22.31 | 81.27 | 86.88 | 89.64 |
| LLaMA 11B | ✓ | **8.37** | **22.37** | **82.53** | **87.96** | **90.64** |
| LLaVA 7B | ✗ | 6.82 | 21.91 | 82.10 | 86.94 | 89.03 |
| LLaVA 7B | ✓ | **7.22** | **21.98** | 81.28 | **86.99** | **89.53** |

Table 5: Ablation study comparing SPRO-prompt method with and without reasoning for two VLMs across three evaluation metrics. Reasoning improves aesthetics (Laion Score), user preference (PickScore), and engagement (EOIG Score) for all buckets.

#### A.2.2 Effect of Training Data Size Across Iterations

We analyze how the size of the training dataset affects the performance of our model across iterations. In particular, we train our model on three different dataset sizes and evaluate the resulting PAP scores. The results demonstrate that using 6K pairs yields the best performance as shown in Table6.

| Training Data size | PAP Score |
|---|---|
| 3k | 22.33 |
| 6k | **22.39** |
| 60k | 22.15 |

Table 6: Performance of models trained at different dataset sizes. Best result obtained with 6K pairs.

These findings suggest that moderate data size with high-quality pairs is more effective than using significantly larger datasets.

### A.2.3  Iterative Refinement Analysis

To explore the potential of prompt-image refinement without additional training, we conducted an iterative feedback loop experiment. The model was prompted with its own generated image and caption to produce refined outputs. This was repeated for two hops to observe runtime improvements.

This experiment focused on difficult samples in the bottom third quartile of initial caption scores. Table 7 summarizes score changes across hops.

| Objective | Metric | Initial Score | Hop 1 | Hop 2 |
|---|---|---|---|---|
| Human Preference | PickScore | 20.31 | +0.73 | +0.84 |
| Aesthetics | LAION Score | 8.18 | +0.10 | +0.13 |
| Engagement | EOIG Score | 77.50 | -0.43 | +1.75 |

Table 7: Runtime improvements on low-scoring samples using iterative refinement.

These results indicate that even without retraining, iterative prompting enables the model to refine suboptimal generations over multiple steps. This has practical potential for human-in-the-loop systems or adaptive image enhancement pipelines.

### A.2.4  Training and Testing data score distributions

The figure 4 illustrates the model's improvement over training iterations. With each iteration of data generation using the training set, the score distribution progressively shifts toward higher values, indicating performance gains. A similar trend is observed on the test set, demonstrating consistent improvement across all three methods: SPRO-Prompt, SPRO-Image, and SPRO-MM.

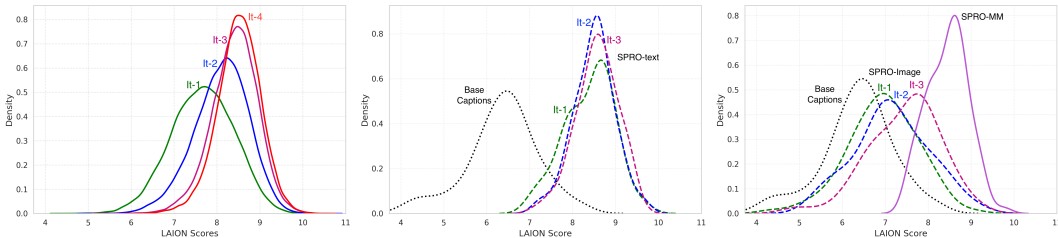

Figure 4: Evolution of self-play scores over training iterations and testing iterations.(i) Displays the LAION score distribution of four iterations of SPRO-text for training data (ii) Displays the LAION score distribution of three iterations of SPRO-text for testing data and Base captions (iii) Displays the LAION score distribution of three iterations of SPRO-image and SPRO-MM. The average scores of generated samples improve rapidly in early iterations but gradually saturate, indicating convergence. The score difference between high- and low-quality samples also narrows, suggesting increasing generation consistency.

### A.2.5   Comparison with CAPO on Aesthetic Quality (VILA Score)

The CAPO paper [12] evaluates aesthetics using the VILA score [8]. To enable a direct comparison, we use the Parti Prompt Dataset [31] and compute VILA scores [8] for images generated by our approach.

| Model | VILA Score |
|---|---|
| CAPO | 6.14 |
| SPRO-Prompt | **6.48** |
| SPRO-Image | 5.95 |
| SPRO-MM | **6.39** |

Table 8: VILA aesthetic scores on Parti prompts using the SPRO methods and CAPO. SPRO-Prompt surpasses CAPO baseline, with other variants showing competitive results.

Our SPRO-Prompt variant outperforms the CAPO baseline, indicating that our method enhances aesthetic quality even on standard prompt benchmarks.

### A.2.6   CLIP Score Sanity Check for SPRO Methods

To ensure the similarity of the generated images relative to the original base images, we computed the CLIP Score [18] similarity between each base image and its corresponding generated image across all evaluation metrics. This sanity check validates that the optimized images remain semantically aligned with the input images while improving on user preference, aesthetics, and engagement by adding details to prompt that do not reflect much semantic change in new image. Table 9 summarizes the mean CLIP Score values for each SPRO method across the three human preference objectives.

| Method | User Preference | Aesthetics | Engagement |
|---|---|---|---|
| SPRO-Prompt | 77.31 | 86.61 | 75.82 |
| SPRO-Image | 79.88 | 88.02 | 78.10 |
| SPRO-MM | 79.12 | 87.44 | 77.34 |

Table 9: Mean CLIP Score similarity between base images and generated images for different SPRO methods, across user preference, aesthetics, and engagement objectives.

### A.2.7   Winrates of prompt-space methods

We evaluate the comparative winrate of prompt space method using different VLM across multiple diffusion models and metrics.

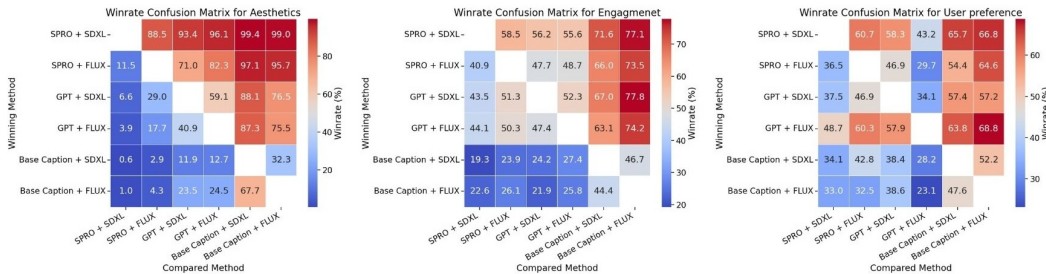

Figure 5: Winrate confusion matrices comparing performance of prompt engineering methods (SPRO-prompt, GPT, Base Caption) paired with diffusion models (SDXL, FLUX) across aesthetic quality (left), user engagement (center), and preference (right) metrics. Values represent percentage frequency with which row methods outperform column methods in direct comparisons.

### A.2.8 Evaluation without Dependence on External Rewards

To examine SPRO's reliance on learned reward models, we conducted an additional experiment using direct human ratings from the AVA dataset as supervision. The goal was to test whether SPRO can align with human aesthetic preferences without any pretrained reward model.

An untrained LLaMA-3.2-11B was used as the Guider VLM. Starting from a base prompt, the model generated three candidate prompts $(p_1, p_2, p_3)$ aimed at improving aesthetics. Instead of scoring generated images with a reward model, each prompt retrieved its top-5 most relevant AVA images using CLIP-based text–image similarity. From these retrieved images, we selected those with the largest difference in human aesthetic ratings to form preference pairs for Direct Preference Optimization fine-tuning.

During evaluation, each test prompt was assigned the highest human aesthetic rating among its retrieved AVA matches. Despite the weak supervision, as shown in Table 10 SPRO achieved a clear improvement 6.80 versus 5.90 for base captions indicating that it can effectively learn aesthetic alignment directly from human annotations.

| Model | Training Method | Human Annotated Score |
|---|---|---|
| Base-Captions | - | 5.90 |
| GPT-4o | Few-shot | 6.20 |
| Llama-3.2-11B | Base | 6.50 |
| Llama-3.2-11B | SPRO iteration-1 | **6.80** |

Table 10: Evaluation of aesthetics using human-annotated scores from AVA dataset without reliance on external reward models.

### A.2.9 Comparison with Strong Reasoning Models on all three Human Preference Objectives

To further evaluate SPRO's generalizability, we compared our framework against advanced reasoning-augmented models, including **Qwen-3**, **GPT-4o**, and **GPT-o3**, across three human preference objectives: aesthetics (LAION Score), user preference (PickScore), and engagement (EOIG Score). Engagement was measured across three difficulty buckets—low, medium, and high—following established evaluation protocols from prior work.

As shown in Table 11, **SPRO-Prompt** consistently outperforms all baselines across all metrics. On the aesthetic objective, it achieves the highest score of **8.60**, surpassing Qwen-3 (7.55), GPT-4o (7.57), and GPT-o3 (7.67). For user preference, SPRO-Prompt reaches 22.42, outperforming GPT-o3 (19.47), GPT-4o (22.31), and Qwen-3 (22.10). Most notably, across all EOIG engagement buckets, SPRO-Prompt attains the best performance: 83.48 (low), 88.36 (medium), and 90.24 (high).

These results demonstrate that SPRO-Prompt surpasses both general purpose and advanced reasoning models in aligning generated outputs with human preference objectives.

| Model | Aesthetics | User Preference | EOIG Score | | |
|---|---|---|---|---|---|
| | | | *Low* | *Mid* | *High* |
| Qwen-3 | 7.55 | 22.10 | 74.12 | 83.12 | 84.23 |
| GPT-4o | 7.57 | 22.31 | 77.70 | 87.83 | 89.95 |
| GPT-o3 | 7.67 | 19.47 | 75.91 | 75.97 | 79.51 |
| LLaMA-3.2-11B | 7.38 | 21.96 | 83.19 | 87.72 | 89.32 |
| SPRO-Prompt (Ours) | **8.60** | **22.42** | **83.48** | **88.36** | **90.24** |
| SPRO-Image | 7.48 | 22.09 | 70.70 | 77.94 | 77.43 |
| SPRO-MM | 8.42 | **22.42** | 83.20 | 87.39 | 89.38 |

Table 11: Performance comparison of SPRO and other reasoning-augmented models across aesthetics(LAION Score), user preference(PickScore), and engagement metrics(EOIG Score).

### A.2.10 Effect of SPRO Iteration on Reasoning-Enabled Models for objective of Aesthetics

Motivated by the strong performance of SPRO-Prompt, we further investigated whether self-play training could enhance the ability of reasoning-based models to generate prompts optimized for aesthetics. Specifically, we employed Qwen-3-7B as the Guider VLM and conducted a single iteration of self-play using the human preference objective of aesthetics. The model generated reasoning chains and corresponding prompt candidates across varying temperature settings. These prompts were used to generate images with a frozen SDXL model, which were then scored using the LAION aesthetic scorer. Based on these scores, preference pairs were constructed and used for fine-tuning via Direct Preference Optimization (DPO), jointly conditioning on reasoning and prompts.

We evaluated the fine-tuned models on the test set using the LAION aesthetic score. As shown in Table 12, SPRO training led to consistent improvements across both reasoning and non-reasoning VLMs. Notably: Qwen-3-7B improved from 7.55 to 8.46 after one iteration, while Llava-7B improved from 6.76 to 7.22 after three iterations and Llama-3.2-11B improved from 7.38 to 8.30.

| Model | Training Method | LAION Score (Aesthetics) |
|---|---|---|
| Base-Captions | - | 6.40 |
| Qwen-3-7B | - | 7.55 |
| Qwen-3-7B | SPRO-it-1 | **8.46** |
| Llava-7B | - | 6.76 |
| Llava-7B | SPRO-it-1 | 7.22 |
| Llama-3.2-11B | - | 7.38 |
| Llama-3.2-11B | SPRO-it-1 | 8.30 |
| Llama-3.2-11B | SPRO-it-3 | **8.60** |

Table 12: Comparison of LAION aesthetic scores before and after SPRO iterations of reasoning-capable models.

### A.2.11 Zero-Shot Evaluation on multiple metrics for same objective

To verify that SPRO-Prompt does not exploit any specific reward function and instead learns generalizable strategies, we conducted zero-shot generalization experiments across multiple metrics of each objective. While SPRO-Prompt was trained using a single reward signal per objective, it was evaluated on unseen reward models representing the same underlying human preference dimension.

For the aesthetic objective, SPRO-Prompt was trained using preference pairs derived from the LAION Aesthetic Score, but evaluated on unseen scorers VILA, the Improved Aesthetic Scorer, and CLIP similarity , using the Flickr dataset. As shown in Table 13, SPRO-Prompt consistently surpasses the base prompt across all metrics, achieving substantial gains on both VILA and the improved aesthetic scorer. These results confirm that SPRO-Prompt generalizes beyond the LAION reward model, capturing a broader notion of aesthetic quality while maintaining semantic alignment as indicated by stable CLIP similarity.

| Model | LAION Score | VILA | Improved Aesthetic | CLIP |
|---|---|---|---|---|
| Base Prompt | 7.38 | 60.74 | 6.30 | **83.21** |
| SPRO-Prompt | **8.60** | **65.16** | **7.47** | 79.50 |

Table 13: Generalization of SPRO-Prompt to multiple aesthetic scorers and CLIP alignment on the Flickr dataset.

For the user preference objective, the model was trained with PickScore as the reward signal and evaluated using ImageReward as an unseen metric on the Pick-a-Pic dataset. As presented in Table 14, SPRO-Prompt outperforms the base prompt across all evaluation metrics—PickScore (22.42 vs. 20.99), ImageReward (1.02 vs. 0.65), and CLIP similarity (89.32 vs. 87.76). This demonstrates that SPRO-Prompt is not overfitted to PickScore but instead learns prompting strategies that generalize across different models of user preference assessment.

| Model | PickScore | ImageReward | CLIP |
|---|---|---|---|
| Base Prompt | 20.99 | 0.65 | 87.76 |
| SPRO-Prompt | **22.42** | **1.02** | **89.32** |

Table 14: Generalization of SPRO-Prompt to multiple user preference scorers and CLIP alignment on the Pick-a-Pic dataset.

Together, these findings show that SPRO-Prompt generalizes across multiple reward models within the same objective class, reinforcing that it optimizes genuine human-aligned quality rather than engaging in reward hacking.

### A.2.12   Multi-Objective Optimization: Aesthetics and CLIP Similarity

To assess SPRO's capability for multi-objective optimization, we extended training to jointly optimize two reward signals: (a) LAION score for aesthetics as the primary objective, and (b) CLIP similarity as a measure of semantic alignment. During self-play, the Guider VLM was conditioned to balance both objectives simultaneously.

As shown in Table 15, SPRO achieves a substantial improvement in aesthetics while maintaining nearly identical CLIP similarity compared to the base prompt. This indicates that SPRO effectively supports multi-objective optimization, enhancing aesthetic quality without compromising text–image semantic consistency.

| Model | LAION Score | CLIP Score |
|---|---|---|
| Base Prompt | 6.40 | 83.19 |
| SPRO-Prompt | **7.90** | **83.28** |

Table 15: SPRO scores for for Multi-objective optimization using LAION aesthetic score and CLIP similarity. SPRO improves aesthetics without compromising image-text alignment.

### A.2.13   Dual Optimization: Aesthetics and User Preference

To evaluate SPRO's ability to optimize multiple human preference objectives simultaneously, we conducted an experiment using two orthogonal signals: aesthetics (LAION Score) and user preference (PickScore). Unlike CLIP similarity, which primarily ensures semantic consistency, these two metrics represent distinct and potentially unaligned preferences. Preference pairs were generated conditioned on both LAION and PickScore, retaining only those where improvements occurred in both metrics. These filtered pairs were used to train the Guider VLM for one iteration, enabling the model to generate reasoning chains and prompts that jointly optimize both objectives.

As shown in Table 16, SPRO-Prompt improves performance across both metrics: LAION Score rises from 6.40 to 7.66, and PickScore increases from 22.69 to 22.96. These results highlight SPRO's capacity to learn prompting strategies that align with multiple non-conflicting human preferences, demonstrating the framework's flexibility in multi-reward scenarios.

| Model | LAION Score | PickScore |
|---|---|---|
| Base Prompt | 6.40 | 22.69 |
| SPRO-Prompt | **7.66** | **22.96** |

Table 16: SPRO scores for for Dual objective optimization of aesthetic quality and user preference. SPRO improves both simultaneously.

### A.2.14 Optimization Under Conflicting Objectives

To investigate SPRO's ability to handle conflicting objectives, we designed a composite pseudo-reward function:

$$\text{Conflicting Score} = 2 \times \text{LAION Score} - 0.5 \times \text{PickScore},$$

which favors images with higher aesthetic appeal while penalizing user preference. This simulates real-world scenarios where maximizing one objective may necessitate compromising another, such as artistic or stylistic choices that enhance aesthetics but reduce broader appeal.

Preference pairs were generated based on improvements in this conflicting score and used to train the Guider VLM for one iteration via SPRO-Prompt. As shown in Table 17, the model successfully increases the LAION aesthetic score from 6.40 to 7.62 while reducing PickScore from 22.69 to 21.70, reflecting the intended trade-off. These results demonstrate that SPRO can effectively optimize for non-aligned or conflicting objectives, highlighting its applicability to complex human preference modeling scenarios.

| Model | LAION Score | PickScore |
|---|---|---|
| Base Prompt | 6.40 | 22.69 |
| SPRO-Prompt | **7.62** | **21.70** |

Table 17: SPRO scores for Conflicting-objective optimization: SPRO increases aesthetics while decreasing user preference as intended by the design of Conflicting reward function.

### A.2.15 Generalization of SPRO across multiple datasets

To further evaluate the robustness and adaptability of SPRO, we tested the trained models on additional datasets containing diverse base prompts and image distributions. This experiment examines whether the prompting strategies learned through self-play generalize effectively across datasets that differ from those used in training, while maintaining strong performance on human preference objectives.

For both the aesthetic and user preference objectives, we conducted evaluations on the PartiPrompts dataset. As shown in Table 18, SPRO-Prompt demonstrates consistent improvements over base prompts across three independent aesthetic reward models: LAION, VILA, and the Improved Aesthetic Scorer. Specifically, SPRO-Prompt improves LAION Score from 6.16 to 8.01, VILA Score from 55.15 to 64.81, and the Improved Aesthetic Scorer from 5.60 to 6.78.

| Method | LAION Score | VILA | Improved Aesthetic |
|---|---|---|---|
| Base Prompt | 6.16 | 55.15 | 5.60 |
| SPRO-Prompt | **8.01** | **64.81** | **6.78** |

Table 18: Evaluation of SPRO-Prompt on the PartiPrompts dataset for the aesthetic objective using multiple reward models.

For the user preference objective, we report results in Table 19, where SPRO-Prompt again achieves a substantial improvement over the base prompt, increasing PickScore from 21.89 to 22.84.

| Method | PickScore |
|---|---|
| Base Prompt | 21.89 |
| SPRO-Prompt | **22.84** |

Table 19: Evaluation of SPRO-Prompt on the PartiPrompts dataset for the user preference objective.

Further, we assess its performance on the LexicaDB [13] dataset—a benchmark used in Reward-Agnostic Prompt Optimization for Text-to-Image Diffusion Models (RATTPO) [10]. This setting allows us to test whether our model, trained on distinct objectives, can effectively generalize to unseen reward functions and datasets.

The RATTPO baseline is explicitly trained to optimize prompts for the ImageReward function, which aligns text-to-image generations with human preference. In contrast, SPRO-Prompt has never been exposed to this reward model during training, providing a true zero-shot generalization test.

Following the RATTPO setup, we randomly sample 160 image–caption pairs from LexicaDB [13] (as the original test set is unavailable). For each instance, we use the base image and prompt as inputs to our user-preference optimized SPRO-Prompt model, generate an improved prompt, and evaluate the resulting generations using the ImageReward metric.

| Model | Approach | Initial ImageReward score | Optimised ImageReward score |
|---|---|---|---|
| RATTPO | Prompt | $0.049 \pm 0.143$ | $1.132 \pm 0.049$ |
| SPRO-Prompt(ours) | Prompt | **0.1588** | **1.439** |

Table 20: Evaluation of SPRO-Prompt on the LexicaDB dataset for the user preference objective.

These results confirm that SPRO generalizes effectively across datasets and reward models, reinforcing its robustness and applicability to diverse prompt–image distributions. The model's ability to maintain improvements on unseen datasets highlights that it learns transferable strategies rather than overfitting to the training distribution.

### A.3    Analysis of reasoning chains generated via Self-play

In this section, we analyze several strategies discovered by our Self-Play framework. These strategies, emerged through iterative optimization via Self-play of intelligent VLM guided by reward score. Using the complete set of data generated through self-play, we categorize the strategies into two groups- those that consistently improve scores and the ones that tend to degrade performance for a specific human preference. This analysis reveals insightful patterns about what types of prompt modifications or stylistic shifts align with specific human preferences as shown in Table 21.

### A.4    Experimental Setup

### A.4.1    Setup

Our SPRO-Prompt framework employs a Guider VLM to generate reasoning chains and optimized image prompts. We use LLaMA-3.2-11B-Vision-Instruct [14] as the backbone for the VLM. All experiments are conducted on 8 NVIDIA A100 GPUs. DPO fine-tuning is performed on same with a per-device batch size of 2, using gradient accumulation to achieve a larger effective batch size. Training is run for 2 epochs using bf16 precision and updates all model parameters.

For image generation, we use a frozen Stable Diffusion XL (SDXL) [17] as the base diffusion model. In the **SPRO-Image** method, we finetune the same SDXL model using high-reward synthetic images paired with their corresponding base captions. We use a resolution of 512×512, a learning rate of 1e-6, the 8-bit Adam optimizer, and train for 50 epochs with a gradient accumulation factor of 4.

In the **SPRO-MM** setup, we combine both components: prompts are generated using the Guider VLM trained via SPRO-Prompt, and images are generated using the diffusion model trained via SPRO-Image.

To assess generalizability of prompts, we also validate our results using two alternative diffusion models: FLUX.1-dev [2] and DALL·E 3 [16].

**Self-Play Data Generation** We generate training data through an iterative self-play procedure involving the Guider VLM and a frozen diffusion model. The process begins with a seed dataset of image-caption pairs tailored to each human preference objective: (1) Aesthetic appeal: 30,000 image-caption pairs from the Flickr30k dataset [30]. (2) Engagement: 35,000 images sampled from the EngagingImageNet train set [9], with equal representation from the top and bottom 20th percentiles of like counts to capture both highly and poorly engaging content.(3) User preference: 17,400 images from the validation split of the Pick-a-pic dataset [11]. Using these seed datasets, the Guider VLM generates prompts via self-play, which are used with the diffusion model to create new image-caption pairs scored by the appropriate reward model. Only samples that exceed the reward

| Human Preference | Focus Area | Winning Strategy | Losing Strategy |
|---|---|---|---|
| User Preference | Language Style | More technical and systematic | More artistic and conceptual |
| | Mood / Lighting | Atmospheric effects grouped with vignette | Atmospheric effects grouped with lighting |
| | Noise and Texture | Noise reduction and texture combined in one strategy | Noise reduction and texture details separated in different strategy |
| | Color and Exposure | Separates contrast and color | Combines color grading + contrast |
| | Detail Work | Sharpening + clarity as a unified function | Detail and texture enhancement separately |
| Aesthetics | Language Style | Artistic and interpretive (e.g., "artistic flourishes," "visual harmony") | Technical and grounded (e.g., "posture," "composition," "depth") |
| | Scene details | Focus on more style and cinematic scenes | Focus on realism and depth in scenes |
| | Composition | Guiding viewers attention through placement | Combine depth of fields and spatial balance |
| | Color Palette | Includes broader scope in colours | Focus on vibrancy keeping natural hues |
| | Persona Point | Digital Artists and Concept Artists | Photographers and Visual Designers |
| Engagement | Strategy focus | Technical enhancements like lighting, motion, and composition | Clear emphasis on visual appeal and connection with audience |
| | Tone | More detailed and descriptive | Straightforward, action-oriented, focusing on engagement tactics |
| | Lighting and Shadows | Focus on background and shadow effects | Focus on dimension and depth |
| | Dynamic elements | Focus on motion blur and dynamic poses | Mentions motion lines or animation |
| | Emotional Connection | Does not directly address emotional connection; focuses more on visual elements and energy through motion. | Directly calls for enhancing emotional connection through facial expressions, body language, and context. |

Table 21: Analysis of Winning and Losing Strategies Across Focus Areas for each Human Preference. These advance stratgies are highly technical , human poreference specific and counter intuitive and hence difficult for humans to explore.

distribution's mean score for base-captions and pass a CLIP-based image similarity threshold are retained.

### A.4.2 Prompts used to generate data via iterative self-play

> **Prompt for Aesthetics**
>
> Think step-by-step and analyze the image to iteratively enhance it. Generate a 60-word diffusion model prompt to produce a version of the image with improved aesthetics. Ensure all original details are preserved, and the enhancements match the style and essence of the original image while improving its visual appeal. **The base caption for the image is:** `caption` **Output format:**
>
> ```
> Step1_Plan: [Specific plan for step-1]
> Step2_Plan: [Specific plan for step-2, building on step-1]
> Step3_Plan: [Specific plan for step-3, building on step-1 & step-2]
> ...
> Caption: [Generated prompt]
> ```
>
> Generate up to six steps and the final prompt. The prompt is extremely important and must not exceed 60 words.

## A.5   Human Study

### A.5.1   Human Evaluation Instructions

We conducted a human evaluation study to assess the quality of synthetic images generated by our pipeline. Below are the exact instructions provided to the participants during the evaluation:

**Instructions:**

- You will be shown 20 prompts.
- Each prompt will be presented with 3 images generated using the same prompt.
- For each prompt, please select the one image you prefer the most.
- You must make a selection to proceed to the next prompt.

### A.5.2   Results from Human Study

To assess alignment with actual human preferences beyond automatic reward signals, we conducted a human evaluation comparing image generations from three prompt types: the original base caption, a GPT-4o-optimized prompt, and our SPRO-prompt. For each input, three images were generated using the same diffusion model (SDXL or Flux), each corresponding to one prompt type. Human annotators (N=20) selected their preferred image in a randomized, blind setting. As shown in Table 22, SPRO-prompt is preferred over GPT-4o in 66.7% of cases and over base captions in 78.9%. In contrast, base captions are chosen over GPT-4o in 52.38% of cases, but only 21.1% when compared to SPRO. These results confirm that prompts optimized via self-play lead to significantly higher human preference.

## A.6   Qualitative analysis

### A.6.1   Caption and Images generated by SPRO-Prompt method

In this section, we present representative examples from the test set across various human preference categories. For each example, we show: the base caption, the image generated by SDXL [17] using the base caption, the SPRO-Prompt i.e our method's refined caption and the image generated by SDXL using the SPRO-Prompt. We observe a consistent improvement in reward scores , significantly

| Comparison | Preferred Candidate | Win Rate (%) |
|---|---|---|
| Base Caption vs. GPT-4o | Base Caption | 52.38 |
| Base Caption vs. SPRO-Prompt | Base Caption | 21.10 |
| SPRO-Prompt vs. Base Caption | SPRO-Prompt | 78.94 |
| SPRO-Prompt vs. GPT-4o | SPRO-Prompt | 66.67 |

Table 22: Pairwise human preference win rates between prompts: base captions, GPT-4o-optimized prompts, and SPRO-optimized prompts. Each row shows the preferred candidate and the percentage of times images generated using the candidate prompt was selected over the alternative.

improvised prompt with more strategic details and notable visual differences between the base and improved images generated. These qualitative improvements further validate the quantitative gains achieved by our method. The samples for user preference is in Table 23, for aesthetics refer Table 24 and for engagement refer Table 25.

### A.6.2 Qualitative results of SPRO-Image and SPRO-MM method

This section presents a qualitative analysis of outputs of SPRO-Image and SPRO-MM method by. Specifically, we compare images generated using original, unmodified captions, images generated by SPRO-Image method i.e a fine-tuned diffusion model that receives the base captions as input and SPRO-MM where the fine-tuned diffusion model is guided by SPRO-Prompt enhanced captions. For the objective of user preference check Table26, for engagement refer Table 27 and for aesthetics refer Table 28.

### A.6.3 Diffusion Model agnostic prompts

We show qualitative samples of images generated using both SDXL and Flux with SPRO-Prompt generated captions. Good reward scores using both models show diffusion agnostic nature of prompts generated. For aesthetics refer Table 29, similarly for objective of user preference refer Table 30 and for objective of enagagement refer Table 31.

### A.7 Broader Impact

Our work contributes to the growing effort of aligning synthetic image generation with human preferences through an explicit alignment framework. This has several positive societal implications. By reducing reliance on large-scale real-world datasets, our approach can alleviate data privacy concerns and promote more ethical and inclusive data practices. Moreover, the ability to generate preference aligned synthetic images could support in multiple applications such as in education to generate more memorable images. However, this capability also introduces ethical challenges. If the alignment process inadvertently captures biased or skewed preferences, it may amplify societal biases or aesthetic norms, particularly in sensitive contexts involving identity, culture, or representation. Care must therefore be taken to audit and evaluate alignment criteria and ensure transparency in the deployment of such models. We encourage future work to further investigate mechanisms for robust preference modeling, fairness-aware alignment, and safeguards against misuse.

### A.8 Dataset Licenses

In this work, we use three available datasets: Flickr30k [30] and Pick-a-Pic [11] and EngagingImageNet [9]. We ensure that all datasets are used in accordance with their respective licenses.

- **Flickr30k Dataset:** The images in the Flickr30k dataset are provided under the Creative Commons Attribution-NonCommercial-ShareAlike 3.0 License (CC BY-NC-SA 3.0).
- **PickAPic Datawset:** The Pick-a-Pic dataset is released under the Creative Commons 1.0 Universal Public Domain Dedication.
- **EngagingImageNet Dataset:** The EngagingImageNet dataset is released under the Creative Commons Attribution 4.0 International License.

| Caption | Caption+SDXL | SPRO-Prompt | SPRO-Prompt + SDXL |
|---|---|---|---|
| A cute very purple cat with purple lavender fur and a cute black top-hat. very cyan blue background |  PickScore: 25.94 | generate a cute purple cat with purple lavender fur and a black top-hat standing on a very cyan blue background, with enhanced lighting, color gradients, and textures, maintaining the original expression and details, while sharpening facial features and adjusting saturation for a vibrant, dreamy atmosphere. |  PickScore: 26.26 |
| An evil villain holding a mini earth |  PickScore: 24.42 | Generate an enhanced image of an evil villain holding a mini earth, with refined facial features, atmospheric lighting, and detailed textures, set in a dark, ominous environment, maintaining the original details and style, with a focus on contrast and color grading to create a more dramatic and sinister mood. |  PickScore: 26.90 |
| bread |  PickScore: 21.24 | Generate a high-quality image of a freshly baked loaf of bread, showcasing its warm golden crust and soft interior. the image should be well-lit, with subtle shadows and highlights. preserve the natural texture and details of the bread. style realistic, food photography. ensure the image is visually appealing and inviting, making the viewer want to take a bite. |  PickScore: 22.89 |
| A Pirate in a Pirateship |  PickScore: 22.15 | Generate a high-quality image of a pirate in a pirateship, preserving original details, with enhanced color tone, contrast, and texture, showcasing the pirate's features, clothing, and the ship's wooden hull, with a warm and inviting atmosphere, as if the pirate is about to set sail, with subtle noise reduction and local adjustments for a refined look. |  PickScore: 23.20 |

Table 23: Comparison of base captions and SPRO-Prompt improvised captions, along with the corresponding images generated by SDXL for higher user prefernce.

| Caption | Caption+SDXL | SPRO-Prompt | SPRO-Prompt + SDXL |
|---|---|---|---|
| A brown-haired man is sitting in front of a sewing machine looking down. |  LAION Score: 7.59 | Vibrant tapestry of life unfolds as a contemplative figure, amidst a kaleidoscope of warm and cool hues, with a golden glow illuminating his introspective face, surrounded by richly detailed fabrics, inviting the viewer into a world of cozy introspection. |  LAION Score: 8.36 |
| caucasian woman in a blue top and black slacks looks off to the side of the camera with a drink in one hand and a amused look upon her face, while a man sits next to her, looking unamused, drink in one hand, straw in the other. |  LAION Score: 7.07 | A vibrant, sunlit scene unfolds with a woman in a radiant blue top, her smile radiant amidst a warm, inviting atmosphere, surrounded by detailed, rustic furniture and lush, detailed brush strokes in the background, where figures mingle, their faces aglow in the soft, golden light, inviting the viewer into a world of joy and connection. |  LAION Score: 8.25 |
| A group of people wearing marathon signs run through a stream.A |  LAION Score: 6.73 | Generate the aesthetic of the runners navigating through a stream, preserving their vibrant colors and detailed expressions while subtly adjusting hues, contrasts, and depth to draw the viewer into the lively scene. create an image where the runners take center stage amidst a harmonious, detailed, and vibrant background. |  LAION Score: 8.16 |
| A black and white dog is swimming through some water. |  LAION Score: 5.62 | a majestic, golden retriever with vibrant fur and playful eyes glides effortlessly through the serene waters, surrounded by a tapestry of blue sky, wispy clouds, and colorful aquatic blooms, all bathed in a warm, golden glow amidst a soft, dreamy palette, inviting the viewer into its whimsical world. |  LAION Score: 7.23 |

Table 24: Comparison of base captions and SPRO-Prompt improvised captions, along with the corresponding images generated by SDXL for better aesthetics.

| Caption | Caption+SDXL | SPRO-Prompt | SPRO-Prompt + SDXL |
|---|---|---|---|
| A person is typing on a laptop keyboard. |  EOIG Score: 33.78 | A person is working on a sleek, modern laptop at a wooden desk, with their hands poised over the keyboard. The screen displays a vibrant desktop wallpaper featuring a stunning mountain landscape reflected in a calm lake during sunset, suggesting a peaceful and focused working environment. |  EOIG Score: 67.38 |
| A french bulldog on a walkboard near a beach |  EOIG Score: 68.16 | Warm whimsical celebration of pets, with pastel colors, soft textures, surrounded by hearts, flowers, and a warm glow, , all set against a clean and uncluttered background, showcasing grateful, expressive face, with a slight texture overlay to add depth and tactility. |  EOIG Score: 92.42 |
| A computer monitor with a game of Overwatch on the screen |  EOIG Score: 71.25 | Compact gaming laptop, big-time performance. ROG Strix GL 12: Upgrade, customize, dominate. Sleek, high-contrast design meets bold, tournament-grade power. Get the edge you need to play like a pro. |  EOIG Score: 74.71 |
| Astronauts posing in front of American flags |  EOIG Score: 14.32 | Revamped Boeing-built Saturn V first stage, astronaut trio: , stand heroically in front of a vibrant, and rocket plume, all set against a warm, nostalgic glow, emphasizing their monumental Apollo 10 mission. . |  EOIG Score: 54.02 |

Table 25: Comparison of base prompts and SPRO-Prompt improvised captions, along with the corresponding images generated by SDXL for better engagement.

| Caption | SDXL | SPRO-Image | SPRO-MM |
|---|---|---|---|
| A curious cat exploring a haunted mansion |  PickScore: 21.47 |  PickScore: 22.20 |  PickScore: 22.87 |
| Woodcut print giant octopus attacks pirate ship, ultra detailed |  PickScore: 21.40 |  PickScore: 23.62 |  PickScore: 23.53 |
| Flower frame symmetrical,on white background pink, purple, peach,cream,blue,vibrant, loose watercolour gouache textured paper naive folk art, whimsical,Lilla Rogers agency illustrator storybook style, subtle patternsstripes |  PickScore: 19.92 |  PickScore: 20.13 |  PickScore: 20.48 |
| Store with a sign that says Stable Diffusion. |  PickScore: 20.76 |  PickScore: 22.60 |  PickScore: 21.47 |

Table 26: Comparison of base captions and corresponding images generated by: (i) SDXL using the base caption, (ii) SPRO-Image, a fine-tuned diffusion model prompted with the base caption, and (iii) SPRO-MM, which uses the fine-tuned model with SPRO-Prompt enhanced prompts. The comparison highlights improvements in image quality and alignment with user preferences.

| Caption | SDXL | SPRO-Image | SPRO-MM |
|---|---|---|---|
| Graceful fishes swimming among blue, blooming water. |  EOIG Score: 92.37 |  EOIG Score: 98.59 |  EOIG Score: 96.64 |
| Man wandering through vast desert landscapes, guided by words. |  EOIG Score: 38.43 |  EOIG Score: 51.17 |  EOIG Score: 52.22 |
| Vibrant meals served on a table. |  EOIG Score: 28.49 |  EOIG Score: 42.73 |  EOIG Score: 60.35 |
| A yellow car is driving down a road. |  EOIG Score: 89.3 |  EOIG Score: 98.59 |  EOIG Score: 96.64 |

Table 27: Comparison of base captions and corresponding images generated by: (i) SDXL using the base caption, (ii) SPRO-Image, a fine-tuned diffusion model prompted with the base caption, and (iii) SPRO-MM, which uses the fine-tuned model with SPRO-Prompt enhanced prompts. The comparison highlights improvements in image quality and higher EOIG score which reflects enagagement.

| Caption | SDXL | SPRO-Image | SPRO-MM |
|---|---|---|---|
| A white dog is splashing through the water. |  LAION Score: 6.91 |  LAION Score: 7.95 |  LAION Score: 8.42 |
| Female fire dancer performing in the middle of a city street with a crowd watching. |  LAION Score: 6.51 |  LAION Score: 8.25 |  LAION Score: 8.78 |
| A black and white dog is swimming through some water. |  LAION Score: 5.62 |  LAION Score: 7.97 |  LAION Score: 7.30 |
| Old guitarist in white clothes performing to an audience. |  LAION Score: 6.57 |  LAION Score: 8.57 |  LAION Score: 8.08 |

Table 28: Comparison of base captions and corresponding images generated by: (i) SDXL using the base caption, (ii) SPRO-Image, a fine-tuned diffusion model prompted with the base caption, and (iii) SPRO-MM, which uses the fine-tuned model with SPRO-Prompt enhanced prompts. The comparison highlights improvements in image quality and higher LAION score which reflects aesthetics.

| Caption | Base Image | SDXL | FLUX |
|---------|-----------|------|------|

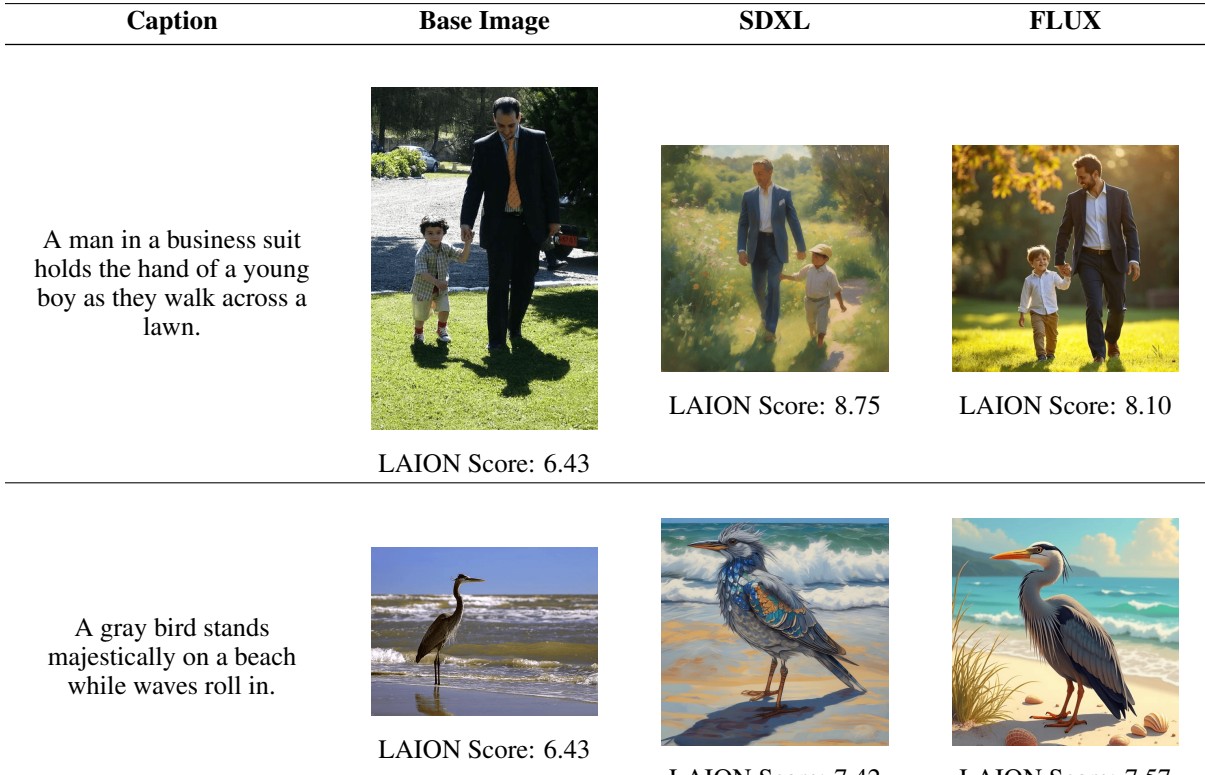

Table 29: Comparison of original image with , images generated using SDXL and FLUX with prompt input generated by SPRO-prompt , for better aesthetics.

| Caption | Base Image | SDXL | FLUX |
|---|---|---|---|
| chanel letterss in the minddle, chanel style, black and white color, decorate design, advertising disign, with water color style red and black poppy flowers around the letters, hand painting, hand water color style, colorful water color background, some blurry poppy flower shapes in background color | 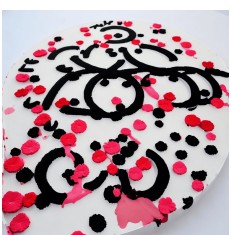
PickScore: 16.65 | 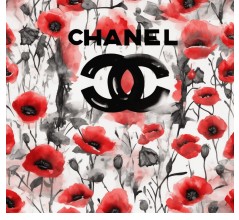
PickScore: 20.67 | 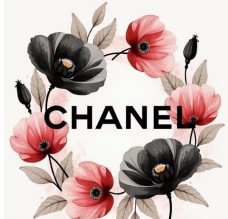
PickScore: 21.40 |
| a cow | 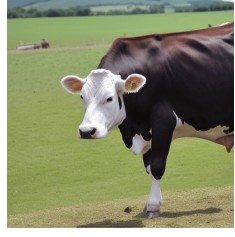
PickScore: 18.35 | 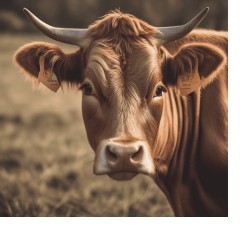
PickScore: 22.79 | 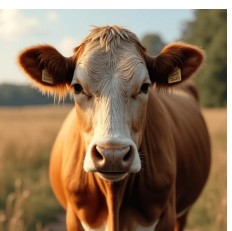
PickScore: 22.91 |

Table 30: Comparison of original image with , images generated using SDXL and FLUX with prompt input generated by SPRO-prompt , for higher user preference.

| Caption | Base Image | SDXL | FLUX |
|---|---|---|---|
| A white airplane flying over the clouds | 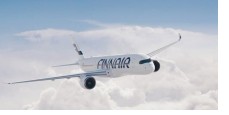
EOIG Score: 76.95 | 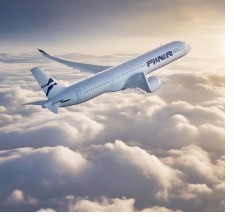
EOIG Score: 98.20 | 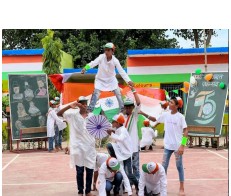
EOIG Score: 95.40 |
| A group of people holding the Indian flag | 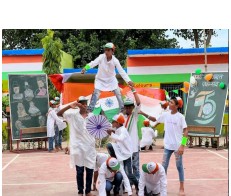
EOIG Score: 53.87 | 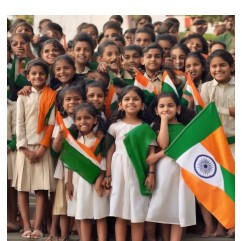
EOIG Score: 72.50 | 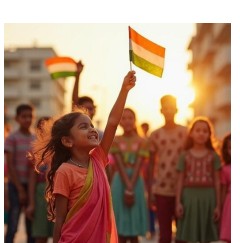
EOIG Score: 92.50 |

Table 31: Comparison of original image with , images generated using SDXL and FLUX with prompt input generated by SPRO-prompt ,for better engagement.

