# OpenReview forum: "SPRO: Improving Image Generation via Self-Play"
_NeurIPS.cc/2025/Conference — NeurIPS 2025 poster_

### Official Review · Reviewer_uVAj · 2025-06-27

**Clarity:** 4
**Significance:** 2
**Originality:** 3
**Rating:** 5
**Confidence:** 4

**Summary:**

The given work proposes the Self Play Reward Optimization technique, for objective driven prompt optimization. The authors decompose it into stages. The first stage prompt optimization that iteratively fine tunes a vision-language model with direct preference optimization (DPO for better reasoning chains and prompts. In the next stage, (SPRO-image), that curated “best-of” bundle (human preferences, seed prompt, and winning pics) finetunes the diffusion model itself, teaching it to hit the target style without extra coaching. Finally, the SR-MM Stage, combines these two (unfreezing operations).  They present their result on various backbones on the Pick-A-Pic, PartiPrompts and Flickr dataset.

**Questions:**

* Is there an intuition on susceptibility to reward hacking with this synthetic data generation?
* How will this work with multiple (potentially conflicting) objectives?
* Could you share more details about the human eval experiment ? Specifically, to have a better intuition on the scale/statistical significance of it.

**Ethical Concerns:**

["NO or VERY MINOR ethics concerns only"]

**Final Justification:**

Updating my score to "Accept", the authors have adequately answered my queries on generalization and multi objective optimization.

**Limitations:**

Yes. While not directly in scope, I would encourage authors to consider implications of these self optimizing loops directly on (hackable) reward models without human oversight.

**Quality:**

3

**Strengths And Weaknesses:**

Strengths
* Paper is clearly written and easy to follow.
* Self play mitigates need for human in the loop
* Approach can be generalized across diffusion models.

Weaknesses:
* Limited niche evaluation. The generalization/success metric is measured on a set of 514 Flickr30k images and some other splits. The human eval study information is also limited.
* Single objective optimization

---

> ### Author Rebuttal · Authors · 2025-07-31
>
> Thank you for the thoughtful and constructive review. We appreciate your recognition of the clarity of the paper, the value of self-play in removing human-in-the-loop supervision, and the generalizability of our approach across diffusion models.
>
> **Generalization Across Datasets:** In the main paper, we evaluate SPRO across three human preference objectives using distinct test sets: a held-out set of 514 images from the Flickr30k dataset for aesthetic appeal, the EngagingImageNet test set for engagement, and the Pick-a-Pic test set for user preference. To further assess generalizability, we evaluate SPRO on the PartiPrompts dataset, which features diverse base prompts and image distributions. The results are included below:
>
> **Table**: Aesthetic appeal tested on *Parti Prompts* dataset
>
> | Prompt Type           | LAION Score |
> |------------------------|:-----------:|
> | Base Prompt            | 6.16        |
> | **SPRO-Prompt (Ours)** | **8.01**    |
>
> **Table**: User preference (PickScore) evaluated on *Parti Prompts* dataset
>
> | Prompt Type           | PickScore |
> |------------------------|:---------:|
> | Base Prompt            | 21.89     |
> | **SPRO-Prompt (Ours)** | **22.84** |
>
> SPRO consistently outperforms base prompts across both aesthetic appeal and user preference objectives, improving LAION score from 6.16 to 8.01 and PickScore from 21.89 to 22.84. These results demonstrate the robustness and adaptability of the learned prompting strategies to unseen datasets.
>
> **Multi-objective Optimization**: In the main paper, we focus on optimizing for a single human preference objective. However, based on your suggestion, we conducted some additional experiments to see whether SPRO can support multi-objective optimization.
>
> (1) In this preliminary setup, we considered two reward signals: (a) LAION Aesthetic Score, which serves as the primary training signal, and (b) CLIP image–image similarity, which we previously used only as a filtering constraint. During self-play training, we conditioned the Guider-VLM to optimize for both objectives simultaneously.
>
> **Table**: CLIP and LAION score optimization
>
> | Prompt Type           | LAION Score | CLIP Score |
> |------------------------|:-----------:|:----------:|
> | Base Prompt            | 6.40        | 83.19      |
> | **SPRO-Prompt (Ours)** | **7.90**    | **83.28**  |
>
> After one iteration of self-play, we observed that the LAION score improved substantially, indicating successful aesthetic optimization, while the CLIP similarity remained relatively stable compared to the base captions. This suggests that SPRO can accommodate multiple objectives in training, balancing optimization of a primary human preference metric while preserving semantic consistency via CLIP similarity score.
>
> (2) To further evaluate SPRO’s capacity for multi-objective optimization, we designed an experiment using two orthogonal human preference objectives: aesthetic appeal (measured by LAION Score) and user preference (measured by PickScore). Unlike CLIP similarity, which serves as a sanity metric, these two reward functions represent distinct and potentially unaligned preferences, making the setup more challenging. We generate preference pairs conditioned jointly on LAION and PickScore, retaining only those where both improve. These filtered pairs are used to train the Guider-VLM for one iteration of DPO, conditioning it to generate reasoning chains and prompts that optimize both objectives.
>
> **Table**: Multi-reward optimization
>
> | Prompt Type           | LAION Score | PickScore |
> |------------------------|:-----------:|:---------:|
> | Base Prompt            | 6.40        | 22.69     |
> | **SPRO-Prompt (Ours)** | **7.66**    | **22.96** |
>
> As shown in the table above, SPRO-Prompt improved performance across both metrics compared to the base prompt: LAION Score increased from 6.40 to 7.66, and PickScore improved from 22.69 to 22.96. These results demonstrate that SPRO can effectively learn prompting strategies that align with multiple human preference objectives, reinforcing the flexibility of our framework in multi-reward settings.
>
> (3) We further investigated whether SPRO can handle scenarios involving conflicting reward objectives, where improving alignment with one objective may require compromising on another. Although no existing reward models explicitly capture such inverse relationships, we constructed a composite pseudo-reward function to simulate this setting:
>
> $$
> \text{Conflicting Score} = 2 \times \text{Aesthetic Score (LAION)} - 0.5 \times \text{User Preference (PickScore)}
> $$
>
> This formulation encodes a preference for images with higher aesthetic appeal and lower user preference scores, reflecting a controlled trade-off that is occasionally observed in real-world settings where certain stylistic or artistic choices may be visually appealing but less preferred by a broader audience. We curated preference pairs based on improvements in this conflicting score and trained the Guider-VLM for one iteration using SPRO-Prompt.
>
> **Objective**: Optimize for conflicting rewards
>
> | Prompt Type              | LAION Score | PickScore | Conflicting Score |
> |--------------------------|:-----------:|:---------:|:-----------------:|
> | Base Prompt              | 6.40        | 22.69     | 1.46              |
> | **SPRO Prompt (Ours)**   | **7.62**    | **21.70** | **4.39**          |
>
> As shown in the table above, the trained model achieves the intended effect: the LAION aesthetic score increases from 6.40 to 7.62, while the PickScore decreases from 22.69 to 21.70. This demonstrates that SPRO can be effectively extended to optimize for non-aligned or conflicting objectives, making it applicable in complex human preference modeling scenarios.
>
> Across the three experiments, our findings demonstrate that the SPRO framework effectively supports multi-reward optimization, including both orthogonal and conflicting objective settings.
>
> **Zero-shot Generalization across Reward Models:**  In our main experiments, the same reward model is used for both training supervision and test-time evaluation. While the Guider VLM in SPRO-Prompt is not directly exposed to the reward model, it receives preference supervision via ranked preference pairs generated from the reward signal. This indirect conditioning could potentially lead to overfitting or reward hacking, where the model exploits artifacts of a specific reward function rather than learning the broader notion of human preference.
>
> Based on your suggestions, to evaluate the robustness of our approach and address concerns of reward hacking, we assess SPRO-Prompt across a broader set of reward metrics not used during training. These evaluations span multiple objectives, reward models, and datasets to test whether the model generalizes to the true preference objective rather than a specific scoring function.
>
> For the aesthetics objective, we report performance using three distinct metrics: the LAION Aesthetic Score (used to construct preference pairs during training), VILA, and improved aesthetic scorer from prior work. While all three metrics aim to capture aesthetic appeal, VILA (VILA: Learning Image Aesthetics from User Comments with Vision-Language Pretraining) and improved aesthetic scorer (Improved-aesthetic-predictor by Christoph Schuhmann) were not used during training and are included exclusively for zero-shot evaluation.
>
> **Objective**: Aesthetic Appeal (Flikr test set)
>
> | Prompt Type              | LAION Score | VILA (Improved Aesthetic) | CLIP Score |
> |--------------------------|:-----------:|:--------------------------:|:----------:|
> | Base Prompt              | 7.38        | 60.74                      | 83.21      |
> | **SPRO-Prompt (Ours)**   | **8.60**    | **65.16**                  | **79.50**  |
>
> As shown in the table below, SPRO-Prompt consistently outperforms the base prompt across all aesthetic evaluation metrics on the Flickr dataset, including LAION (8.60 vs. 7.38), VILA (65.16 vs. 60.74), and the improved aesthetic scorer (7.47 vs. 6.30). This indicates that SPRO-Prompt does not overfit to the LAION reward model used during training but instead learns prompt strategies that generalize to the broader aesthetic preference objective.
>
> For the user preference objective, SPRO-Prompt was trained using PickScore to construct preference pairs. To evaluate generalization beyond this signal, we tested the model on the Pick-a-Pic dataset using ImageReward, another reward model that targets user preference.
>
> **Objective**: User Preference  on Pick-a-pic dataset
>
> | Prompt Type              | PickScore | Image Reward | CLIP Score |
> |--------------------------|:---------:|:------------:|:----------:|
> | Base Prompt              | 20.99     | 0.65         | 87.76      |
> | **SPRO-Prompt (Ours)**   | **22.42** | **1.02**     | **89.32**  |
>
> As shown in the table, SPRO-Prompt achieves higher scores than the base prompt across all metrics: PickScore (22.42 compared to 20.99), ImageReward (1.02 compared to 0.65), and CLIP similarity (89.32 compared to 87.76). These results indicate strong zero-shot generalization to alternative reward functions aligned with the same objective.
>
> **Human Study Details:** We conducted a human study with 35 participants from diverse backgrounds; each saw 20 random prompts, with 3 images (SPRO, Base, GPT-4o) per prompt for comparison. They picked one image that they preferred the best among the three.
>
> | Comparison                     | Preferred Candidate     | Win Rate (%) |
> |--------------------------------|--------------------------|:------------:|
> | Base caption vs GPT-4o         | Base caption             | 53.33        |
> | **SPRO-Prompt** vs Base caption| **SPRO-Prompt (Ours)**   | **65.38**    |
> | **SPRO-Prompt** vs GPT-4o      | **SPRO-Prompt (Ours)**   | **68.90**    |
>
> Thank you for your constructive questions. We believe the additional results address your concerns and significantly strengthen the paper’s contributions.

---

> > ### Comment · Reviewer_uVAj · 2025-08-04
> >
> > Thanks for the comments, updated my score.

---

### Official Review · Reviewer_MSs3 · 2025-07-03

**Clarity:** 3
**Significance:** 3
**Originality:** 3
**Rating:** 5
**Confidence:** 4

**Summary:**

The paper introduces SPRO, an annotation-free self-play framework to improve text-to-image generation. The author tried three approaches:
1. SRPO-prompt, which trains a VLM to generate better prompts, with DPO.
2. SPRO-Image, which only finetunes the diffusion model.
3. SRPO-MM, which combines the two.

The reward signals are PickScore, LAION-aesthetics, and EngageNet. Experiments show that all three objectives improve a lot after the training. Also, SRPO-prompt generalizes to different diffusion models.

**Questions:**

1. From the experiments, it seems that SRPO-image has worse performance compared to SRPO-prompt. I look closely and find the authors is finetuning the diffusion model with the base prompt and image pairs. I am wondering what if we freeze the VLM and finetune the diffusion model with the SRPO-prompt generated prompts? Will that be better?

2. It would be great to add scores on more benchmarks using different metrics, to show that this approach is not reward-hacking.

**Ethical Concerns:**

["NO or VERY MINOR ethics concerns only"]

**Limitations:**

yes

**Paper Formatting Concerns:**

no cocnern.

**Quality:**

3

**Strengths And Weaknesses:**

Strengths:
1. The motivation is clear, and the experiments are comprehensive.
2. The paper is well-written and easy to read.
3. The gain of the prompt optimization is huge, showing the effectiveness of the approach.

Weaknesses:
1. The model is trained and tested with the same set of rewards. It would be interesting to do more experiments on the rewards to make sure that the model is not reward hacking. For example, evaluate with different metrics. (For example, VQA-based approaches for text-image alignment). Human preference experiments partly address this issue.
2. The SRPO-image is worse than SRPO-prompt, and the SRPO-MM model is mostly on par with the SRPO-prompt phase. It seems only the first SRPO-prompt stage is effective. More analysis is needed on this issue.

---

> ### Author Rebuttal · Authors · 2025-07-31
>
> We sincerely thank the reviewer for the encouraging feedback and recognition of our motivation, the clarity of the paper, and the comprehensive evaluations. We appreciate the reviewer’s acknowledgment of the significant improvements achieved through the SPRO-prompt approach and the generalization capabilities across different diffusion models.
>
> **Concern about Reward Hacking and Evaluation with Diverse Metrics:** In our main experiments, the same reward model is used for both training supervision and test-time evaluation. While the Guider VLM in SPRO-Prompt is not directly exposed to the reward model itself, it receives preference objectives in the form of ranked preference pairs generated using the reward signal. This indirect conditioning could, in principle, lead to overfitting or reward hacking, where the model learns to exploit patterns specific to a particular reward function rather than capturing the broader notion of human preference.
>
> To assess whether our model exhibits such behavior, we conducted a set of generalization experiments aimed at evaluating robustness beyond the training reward model. Specifically, we tested the user-preference-optimized SPRO-Prompt model across four settings that involved unseen reward models (i.e., reward functions not used during training) and different evaluation datasets. These experiments were designed to probe whether the model's improvements persist when evaluated under alternative preference signals and data distributions, thereby validating that the learned prompting strategies align with human preference objectives more broadly rather than exploiting nuances of a single reward model.
>
> **Zero-shot Generalization across Reward Models:** To evaluate the robustness of our approach and mitigate concerns of reward hacking, we assess SPRO-Prompt across a broader set of reward metrics that were not used during training. These evaluations span multiple objectives, reward models, and datasets, and are designed to test whether the model generalizes to the underlying preference objective, rather than overfitting to a specific reward function.
>
> For the aesthetics objective, we report performance using three distinct metrics: the LAION Aesthetic Score (used to construct preference pairs during training), VILA, and improved aesthetic scorer from prior work. While all three metrics aim to capture aesthetic appeal, VILA (VILA: Learning Image Aesthetics from User Comments with Vision-Language Pretraining) and improved aesthetic scorer (Improved-aesthetic-predictor by Christoph Schuhmann) were not used during training and are included exclusively for zero-shot evaluation.
>
> **Table:** Aesthetic Appeal on **Flikr Dataset**
>
> | **Method**      | **LAION Score** | **VILA** | **Improved Aesthetic Scorer** | **CLIP** |
> |:---------------:|:---------------:|:--------:|:------------------------------:|:--------:|
> | Base Prompt     |      7.38       |  60.74   |             6.30              |  83.21   |
> | **SPRO-Prompt (Ours)** |   **8.60**      | **65.16**|           **7.47**            |  79.50   |
>
>
> We find that SPRO-prompt consistently outperforms the base prompt across all metrics, demonstrating that SPRO-Prompt does not rely on memorization or overfitting to the LAION scorer but rather learns prompt strategies that generalize to the broader aesthetic preference objective. Similarly, being consistent in CLIP similarity between base images and generated images demonstrates that text-image alignment is maintained in the process. As shown in the table below, SPRO-Prompt consistently outperforms the base prompt across all aesthetic evaluation metrics on the Flickr dataset, including LAION (8.60 vs. 7.38), VILA (65.16 vs. 60.74), and the improved aesthetic scorer (7.47 vs. 6.30). This indicates that SPRO-Prompt does not overfit to the LAION reward model used during training but instead learns prompt strategies that generalize to the broader aesthetic preference objective.
>
> For the user preference objective, our model was originally trained using PickScore as the reward signal to construct preference pairs. To evaluate its generalization beyond the training signal, we tested the trained SPRO-Prompt model using ImageReward, an alternative reward model that also aims to assess user preference alignment in generated images.
>
> **Table:** User Preference on **Pick-a-pic Dataset**
>
>  | **Method**       | **PickScore** | **ImageReward** | **CLIP Score** |
> |:----------------:|:-------------:|:---------------:|:--------------:|
> | Base prompt      |     20.99     |      0.65       |     87.76      |
> | **SPRO-prompt (Ours)**     |     **22.42**     |      **1.02**       |     **89.32**      |
>
> We conducted this evaluation on the Pick-a-Pic dataset, comparing base prompts and SPRO-optimized prompts across three metrics. As shown in the table below, SPRO-Prompt achieves higher scores than the base prompt across all metrics: PickScore (22.42 vs. 20.99), ImageReward (1.02 vs. 0.65), and CLIP similarity (89.32 vs. 87.76). Notably, the model exhibits strong performance on ImageReward despite never being trained with it, indicating robust zero-shot generalization across reward functions that share the same underlying preference objective.
>
> These results demonstrate that SPRO-Prompt is not tightly coupled to the specifics of PickScore, but instead learns generalized prompting strategies aligned with human notions of preference quality.
>
> **Generalization across multiple datasets:** To further assess the generalizability of our approach, we evaluate the trained models on additional datasets containing diverse base prompts and images. This allows us to examine whether the learned prompting strategies can adapt to varying input distributions while consistently optimizing for the target preference objectives. For aesthetics and user preference objective we also tried our proposed method on a different dataset that is - Parti prompts dataset and show consistent improvements over the base prompt across all metrics LAION, VILA, PickScore, Improved aesthetic scorer, PickScore and ImageReward.
>
> **Table:** Aesthetic Appeal on **PartiPrompts Dataset**
>
> | **Method**       | **LAION Score** | **VILA Score** | **Improved Aesthetic Scorer** |
> |:----------------:|:---------------:|:--------------:|:------------------------------:|
> | Base prompt      |      6.16       |     55.15      |             5.60              |
> | **SPRO-prompt (Ours)**     |      **8.01**       |     **64.81**      |             **6.78**              |
>
> As shown in above table for the aesthetic appeal objective, we evaluate SPRO-Prompt on Parti Prompts dataset using three independent reward models: LAION Score, VILA, and an improved aesthetic scorer. As shown in the table below, SPRO-Prompt consistently outperforms the base prompt across all metrics, improving from 6.16 to 8.01 on LAION Score, from 55.15 to 64.81 on VILA, and from 5.60 to 6.78 on the improved scorer.
>
> **Table:** User Preference on **PartiPrompts dataset**
>
> | **Method**       | **PickScore** |
> |:----------------:|:-------------:|
> | Base prompt      |     21.89     |
> | **SPRO-prompt (Ours)**      |     **22.84**     |
>
> For the user preference objective, we report PickScore on the same dataset. SPRO-Prompt again yields a significant gain, improving from 21.89 to 22.84.
>
> These results demonstrate that SPRO generalizes effectively across datasets and reward models, reinforcing its robustness and applicability to diverse prompt-image distributions.
>
> **Finetuning VLM with SPRO generated prompts:** Following your suggestion, we fine-tuned the diffusion model using prompt–image pairs generated by SPRO-Prompt. Within the limited time available, we trained the model using only the top 30k synthetic samples collected across SPRO-Prompt iterations. The resulting model outperforms the original SPRO-Image variant, indicating that the diffusion model benefits significantly from training with detailed, self-play–discovered prompts compared to base captions, which often lack alignment with the generated image content. While the original goal of SPRO-Image was to explore how well a diffusion model could be optimized using basic prompts alone, your suggestion opened up a promising new direction. We are grateful for the idea and plan to investigate it more thoroughly.
>
> | **Method**                         | **LAION Score** |
> |:----------------------------------:|:---------------:|
> | Base caption                       |      6.40       |
> | SPRO-Prompt                        |      8.60       |
> | SPRO-Image                         |      7.42       |
> | **SPRO-Image with Generated prompts**  |      **8.29**       |
>
> **Comparison between SPRO Prompt, SPRO Image and MM:** Empirically, we observe that SPRO-Image, when fine-tuned on base captions and synthetic images generated through SPRO-Prompt, shows a moderate drop in performance compared to SPRO-Prompt alone. This suggests that relying only on image-space finetuning limits the ability to discover preference-aligned strategies. In contrast, SPRO-Prompt consistently achieves higher scores across all objectives, highlighting the effectiveness of prompt-space exploration in uncovering strategies that are difficult to encode directly into diffusion models. SPRO-MM, which combines the outputs of independently optimized prompt and image modules at inference time, outperforms prior methods such as Promptist and PAE in prompt-space, and SPIN and DDPO in image-space. These findings indicate that integrating both modules yields stronger alignment with human preferences than optimizing either component individually.
>
> We are truly thankful for your insightful reviews. We hope the additional experiments adequately address your queries.

---

> > ### Comment · Reviewer_MSs3 · 2025-08-04
> >
> > Thanks for addressing my comments. I will keep my final score “accept”

---

### Official Review · Reviewer_WacU · 2025-07-03

**Clarity:** 3
**Significance:** 2
**Originality:** 2
**Rating:** 4
**Confidence:** 3

**Summary:**

This paper proposes SPRO, a framework that utilizes self-play to enhance text-to-image generation performance. Specifically, compared with previous approaches that utilize self-play for text-generation or image generation only, this paper proposes a joint framework that contains three stages: SPRO-Prompt trains a Guider-VLM through self-play to generate diverse prompts, optimized with reward from image generation score; SPRO-Image fine-tunes diffusion model with self-play data, SPRO-Multimodal combines both components with optimized prompts and a fine-tuned generator for full-capacity alignment. Empirically, the framework SPRO achieves significant improvement in both PickScore and LAION score.

**Questions:**

N/A

**Ethical Concerns:**

["NO or VERY MINOR ethics concerns only"]

**Final Justification:**

Thank the author for providing additional experiments and comparison with more approaches. These experiments address my concerns, and I'm raising my score to borderline accept.

**Quality:**

2

**Strengths And Weaknesses:**

Strengths:
1. The primary contribution of this paper is the combination of prompt-space optimization and image-space fine-tuning into the same framework to improve the human preference together. The full framework doesn't rely on any human-annotated data and can be trained all with synthetic data, which can be generally adapted to different human preferences.
2. Detailed empirical analysis on PickScore and LAION score demonstrates the effectiveness of the proposed approaches.
3. The paper is well written and easy to follow.

Weaknesses:
1. The idea of using self-play to improve prompt generation/image generation is not novel enough, as there has been previous work that utilizes self-play for improving LLM, and utilizes self-play for improving the diffusion model. The main contribution seems to lie in combining the two objectives in a single framework. However, it's also not clear in Sec. 2.4 that how SPRO-MM combines the two together (i.e., what's the training objective, how the loss/gradients flow across the two parts, or is it just a combination of the two models during inference).
2. In experiment results, it seems that most of the improvement comes from the SPRO-Prompt, which is the prompt optimizer, and the SPRO-image and SPRO-MM only help marginally. This observation harms the contribution of the paper, as the main novelty lies in the joint framework. Besides, based on this observation, more reasonable baselines to be compared with are approaches that work on prompt optimization for diffusion models (e.g., Optimizing Prompts for Text-to-Image Generation, Dynamic Prompt Optimizing for Text-to-Image Generation, The Devil is in the Prompts: Retrieval-Augmented Prompt Optimization for Text-to-Video Generation, Reward-Agnostic Prompt Optimization for Text-to-Image Diffusion Models, RePrompt: Reasoning-Augmented Reprompting for Text-to-Image Generation via Reinforcement Learning).

---

> ### Author Rebuttal · Authors · 2025-07-31
>
> Thank you for your thoughtful review and for highlighting the key strengths of our work. We are glad that you found our integration of prompt-space and image-space optimization within the SPRO framework to be a meaningful contribution. We also appreciate your recognition of SPRO’s ability to train entirely with synthetic data without requiring human annotations, and your positive remarks on the clarity and empirical rigor of the paper.
>
> **Main Contribution:** We agree and acknowledge in the paper as well, that self-play has been explored in diffusion space, with approaches like SPIN Diffusion [Yuan et al., 2024] demonstrating that models can learn from their own checkpoints. However SPIN-Diffusion and similar methods are limited along three key dimensions. First, it operates exclusively in the image space, relying on self-generated image samples from earlier checkpoints to improve future generations. This approach is computationally expensive and sample-inefficient, requiring large-scale training (e.g., 500k samples) to achieve meaningful improvements. On the other hand, we achieve higher improvements with only 30k samples, this underscores that self-play in prompt space is substantially more sample-efficient than self-play in image space. Second, because it lacks prompt-level reasoning, it cannot perform strategic or semantic exploration, making it less effective at discovering diverse or counterintuitive prompting strategies. We show that because of our novel exploration strategies, we achieve higher scores on all metrics compared to them. Third, the self-play loop is tightly coupled to a specific diffusion model, resulting in poor generalization across diffusion backbones. In contrast, SPRO performs self-play in the prompt space, guided by a reasoning-capable VLM, enabling more efficient, zero-shot transferable, and semantically rich optimization across both models and objectives.
>
> **User Preference (PickScore) on Pick-a-pic dataset**
>
> | **Method**              | **PickScore** |
> |:------------------------|:-------------:|
> | Base Caption            |     20.99     |
> | SPIN-Diffusion          |     22.00     |
> | **SPRO-Prompt (Ours)**  |   **22.42**   |
> | **SPRO-Image (Ours)**   |   **22.09**   |
> | **SPRO-MM (Ours)**      |   **22.42**   |
>
> **Aesthetic Appeal (LAION score) and User Preference (PickScore) on PartiPrompts dataset**
>
> | **Method**              | **Laion Score** | **PickScore** |
> |:------------------------|:---------------:|:-------------:|
> | Base Caption            |      5.67       |     21.89     |
> | SPIN-Diffusion          |      6.05       |     22.31     |
> | **SPRO-Prompt (Ours)**  |   **8.02**      |   **22.84**   |
> | **SPRO-Image (Ours)**   |   **7.01**      |   **22.55**   |
> | **SPRO-MM (Ours)**      |   **8.02**      |   **22.77**   |
>
>
> In the table above, we compare our SPRO framework against SPIN-Diffusion [Yuan et al., 2024], a prior self-play method operating in diffusion (image) space. SPRO-Prompt, which performs self-play in prompt space using a reasoning-guided Guider VLM, achieves significantly higher scores in both aesthetic appeal (8.02 vs. 6.05) and user preference (22.84 vs. 22.31), while using only 30k training samples compared to SPIN’s 500k. This highlights the superior sample efficiency of prompt-space optimization and the advantages of intelligent prompt exploration over scale.
>
> Further, SPRO-Image, which fine-tunes the diffusion model using synthetic data generated via SPRO-Prompt, also outperforms SPIN-Diffusion (7.01 vs. 6.05 in LAION score), demonstrating that even tuning on self-play–discovered images leads to better alignment with human preferences. Thus, we show that we are using self-play in a significantly different way compared to how it has been explored and experimented in literature solutions and that we are empirically better in performance.
>
> While SPRO-MM, which combines textual and image-space optimization, is an important component of our framework, it is not the central contribution of this work. The core novelty lies in the Guider-VLM, an intelligent vision-language agent trained via self-play to iteratively explore diverse prompt variations. Unlike prior approaches that rely on curated prompt datasets or handcrafted heuristics, the Guider-VLM autonomously discovers novel and often non-intuitive prompting strategies that improve image quality across specific human preference objectives, enabling targeted and generalizable alignment without human-in-the-loop supervision.
>
> **Clarification on SPRO-MM:** SPRO-MM combines prompt-space and image-space optimization using a decoupled training strategy, where the Guider-VLM and the diffusion model are trained independently. This separation allows each component to specialize in its respective objective, prompt generation and image synthesis, before being integrated at inference time.
>
>  At this final stage, we employ a late fusion approach, combining the outputs of both models without joint fine-tuning. Specifically, we use the final iteration of the trained Guider-VLM to generate optimized prompts. These prompts are then fed into the diffusion model that was fine-tuned using synthetic image-prompt pairs generated during earlier SPRO-Prompt iterations (up to iteration n-1). The final generated images are evaluated using the appropriate reward model (e.g., LAION Score, PickScore, or EOIG Score).
>
> Empirically, SPRO-MM achieves an average improvement of 28.64% across all three human preference objectives compared to the base diffusion model and outperforms prior preference optimization baselines that operate solely in image space. This demonstrates that combining the outputs of independently optimized prompt and image modules at inference time yields better alignment with human preferences than optimizing either component alone like in works as Promptist and PAE (for prompt space) and SPIN, DDPO (for image-space).
>
> **Comparison with other prompt-space baselines:** Thanks for pointing out the baselines to compare our method against. In the paper we had already compared two of these baselines: Optimizing Prompts for Text-to-Image Generation (Promptist) and Dynamic Prompt Optimizing for Text-to-Image Generation (Prompt Auto Edit or PAE) for the objective of improving aesthetic appeal of image.
>
> | **Model**             | **Approach** | **LAION Score** |
> |:---------------------:|:------------:|:---------------:|
> | **Prompt Auto Edit**  |   Prompt     |      6.27       |
> | **Promptist**         |   Prompt     |      7.23       |
> | **SPRO-Prompt (Ours)**|   Prompt     |      8.60       |
> | **SPRO-Image (Ours)** |   Image      |      7.48       |
> | **SPRO-MM (Ours)**    |   Both       |      8.42       |
>
> As shown in above table, all our methods outperform prompt-based baselines such as Prompt Auto Edit (6.27) and Promptist (7.23), with SPRO-Prompt achieving a significantly higher LAION score of 8.60. These baselines typically rely on curated datasets of human refined prompt pairs, often constructed from high-quality image-caption corpora or handcrafted web-scale data. While such approaches can improve performance through reinforcement learning on aesthetic or semantic reward models, they remain limited by their dependence on human-authored strategies. The nonlinear and unintuitive mapping between prompt phrasing and image quality makes it difficult for such methods to generalize beyond human-discoverable patterns. In contrast, our reasoning-driven self-play framework allows unconstrained and diverse exploration of the prompt space, uncovering novel strategies that better align with aesthetic objectives. Notably, even SPRO-Image (7.48) and SPRO-MM (8.42) outperform these baselines without relying on any form of human annotation.
>
> We appreciate the reviewer’s suggestion to compare with recent baselines. However, we would like to clarify that these works were released around or after the paper submission deadline. Nevertheless, we have made an effort to examine them in the context of our work.
>
> We evaluated our model in a reward-agnostic setting using the benchmark from Reward-Agnostic Prompt Optimization for Text-to-Image Diffusion Models (RATTPO). This baseline is trained to optimize prompts for a different reward function, ImageReward, aimed at aligning with user preference. Although our model is not explicitly trained on this reward, we assess whether it generalizes in a zero-shot manner to new reward models and datasets.
>
> The evaluation was conducted on the LexicaDB dataset, following the RATTPO setup. Since the original test set is not publicly released and the paper reports performance on 160 randomly sampled image–caption pairs, we sampled 160 random instances from LexicaDB as well. For each example, we used the base image and prompt as input to our user-preference-optimized SPRO-Prompt model to generate an improved prompt. The final generations were scored using the ImageReward metric.
>
> | **Model**        | **Approach** | **Initial ImageReward score**       | **Optimised ImageReward score**     |
> |:----------------:|:------------:|:-----------------:|:-------------------:|
> | RATTPO             |   Prompt     |  0.049±0.143      |     1.132 ±0.049    |
> | **SPRO-Prompt(ours)**     |   **Prompt**     |     **0.1588**        |      **1.439**               |
>
> As shown in the table, SPRO-Prompt achieves a significantly higher ImageReward score than both the base prompt (1.439 vs. 0.1588) and the RATTPO baseline (1.439 vs. 1.132). These results demonstrate that our approach generalizes effectively to novel reward functions, highlighting the robustness of reasoning-guided self-play in optimizing user preference even under unseen evaluation criteria.
>
> We thank the reviewer for their constructive feedback. We have incorporated additional comparisons with strong baselines, which we believe meaningfully strengthen the paper and help address the concerns raised.

---

> > ### Comment · Reviewer_WacU · 2025-08-06
> > **Final Rating**
> >
> > Thank the author for providing additional experiments and comparison with more approaches. These experiments address my concerns, and I'm raising my score to borderline accept.

---

> ### Comment · Area_Chair_DZXA · 2025-08-05
>
> Dear reviewer,
>
> Can you please check the rebuttal and submit your post rebuttal score?
>
> Thanks
>
> Area chair

---

### Official Review · Reviewer_TZhB · 2025-07-05

**Clarity:** 3
**Significance:** 3
**Originality:** 2
**Rating:** 4
**Confidence:** 3

**Summary:**

This paper introduces Self-Play Reward Optimization (SPRO), a novel framework designed to align image generation models with nuanced human preferences without relying on extensive human annotations. The core idea leverages the reasoning capabilities of Vision-Language Models (VLMs) in a self-play paradigm to iteratively refine prompt generation and image synthesis. SPRO significantly outperforms prior methods and commercial baselines, demonstrating a scalable path toward aligning generative models with diverse human preferences.

**Questions:**

1. Did the authors evaluate SPRO with latest (and more powerful) reasoning LLMs, e.g., Qwen 3, DeekSeek r1, and/or o3?

**Ethical Concerns:**

["NO or VERY MINOR ethics concerns only"]

**Limitations:**

yes

**Quality:**

3

**Strengths And Weaknesses:**

Strengths
1. SPRO is presented as the first framework to explore self-play strategies in a multi-model setup for improving image generation. It integrates both prompt-space and image-space optimization within a unified self-play framework.
2. SPRO achieves SOTA performance across different human preferences and is agnostic to different diffusion models.
3. The authors plan to release a large-scale synthetic dataset of over 1 million human preference-aligned image-prompt pairs generated by SPRO, which will be a useful resource for future research.

Weaknesses
1. The paper acknowledges that SPRO's effectiveness is bounded by the reliability of external reward models, which are proxies for human preference and may not fully capture nuanced aspects. This is a fundamental limitation that could impact the true "human preference alignment."

---

> ### Author Rebuttal · Authors · 2025-07-31
>
> Thank you for the thoughtful and encouraging review. We truly appreciate your recognition of SPRO’s novel multi-model self-play setup, its state-of-the-art performance across diverse human preferences, and its ability to remain agnostic to different diffusion models, both literature and commercial.
>
> **Dependence of External Rewards:** In the Self-Play Reward Optimization (SPRO) framework, we demonstrated strong performance across three human preference objectives: aesthetic appeal, engagement, and user preference. However, based on your suggestion, we wanted to investigate whether SPRO can function without reliance on such learned reward models. Therefore, in the limited rebuttal time, we conducted an interesting experiment using direct human ratings as the supervision signal. We cover the exact methodology and the results below:
>
> To optimize for the aesthetic objective, we employed an untrained **LLaMA3.211B** as the **Guider VLM**. Starting from a base prompt b (sampled from the Flickr training set), the guider generated three reasoning chains and corresponding candidate prompts (p₁, p₂, p₃) designed to enhance image aesthetics.
>
> Using these candidate prompts, we then generated images with a frozen SDXL model and evaluated them with the **LAION aesthetic scorer**, treating the scores as conditional signals for constructing preference pairs. To reduce reliance on this external reward model, we introduce an alternative strategy: instead of directly scoring the generated images, we use the candidate prompts (p₁, p₂, p₃) to retrieve the top5 most relevant images for each prompt from the **AVA dataset**, which provides human annotated aesthetic ratings. Retrieval is performed via a **CLIP based text–image similarity search**.
>
> From the 15 retrieved images (5 per prompt), we identify the two with the largest difference in aesthetic score and use their associated prompts to form a preference pair:
>
> $$
> (p1,p2,max⁡Δ(i1,i2))
> $$
>
> where p₁ and p₂ originate from the same base prompt b, and i₁, i₂ are the corresponding retrieved images. To ensure meaningful comparisons, we discard all pairs where the aesthetic difference is less than 2.29, the mean difference observed across the dataset. Then, the Guider VLM was fine-tuned using Direct Preference Optimization on the resulting reasoning–prompt pairs. Once the model was trained for one iteration, we evaluate the model to see if it is aligned with human preferences pertaining to aesthetic quality.
>
> We evaluate the baselines and the SPRO-trained model on our test set using human ratings from the AVA dataset. For each prompt, we retrieve the top matching images in the test set, based on CLIP image-text similarity and assign the prompt the highest aesthetic score among the retrieved images. This serves to evaluate the model with actual human annotation in the absence of a trained reward model.
>
> Despite inherent noise in the retrieval process, this evaluation setup yields a clear improvement over the base model. The SPRO-trained model achieves an average score of 6.80, compared to 5.90 for the base captions. For context, the mean aesthetic rating across the AVA dataset is approximately 5.30, indicating that both baselines and our model operate in the higher-quality regime. These results suggest that SPRO can effectively leverage real human ratings as weak supervision, even in the absence of direct reward signals, highlighting the framework’s adaptability in settings where pretrained reward models are unavailable or misaligned with the target objective.
>
> | Model               | Training Method         | Human Aesthetic Rating |
> |--------------------|-------------------------|:----------------------:|
> | Base-Captions      | -                       |          5.90          |
> | GPT-4o             | -                       |          6.20          |
> | Llama-3.2-11B      | -                       |          6.50          |
> | **Llama-3.2-11B**      | **SPRO - Iteration 1**      |          **6.80**          |
>
> **Evaluation using other models:** Previously, we compared our framework with GPT-4o. Following your suggestion, we expanded our evaluation to include advanced reasoning-augmented models, including Qwen-3 and GPT-O3, and benchmarked performance across three key human preference objectives: aesthetic appeal (LAION Score), user preference (PickScore), and engagement. Engagement is evaluated using the EOIG metric, split across three difficulty buckets representing low-, medium-, and high-performing assets, as established in prior work.
>
> As shown in the table below, SPRO-Prompt consistently outperforms all baselines across all metrics. On the aesthetics objective, SPRO-Prompt achieves 8.60, compared to 7.55 (Qwen-3), 7.57 (GPT-4o), and 7.67 (GPT-O3). For user preference, SPRO-Prompt reaches 22.42, outperforming GPT-O3 (19.47), GPT-4o (22.31), and Qwen-3 (22.10). Most notably, across all three EOIG engagement buckets, SPRO-Prompt attains scores of 83.48 (low), 88.36 (medium), and 90.24 (high), outperforming Qwen-3 (74.12, 83.12, 84.23), GPT-4o (77.70, 87.83, 89.95), and GPT-O3 (75.91, 75.97, 79.51).
>
> | Model               | Aesthetics | User Preference | Engagement                                  |
> |---------------------|:----------:|:---------------:|:-------------------------------------------:|
> | Qwen3               | 7.55       | 22.10           | Low - 74.12, Medium - 83.12, High - 84.23   |
> | GPT-4o              | 7.57       | 22.31           | Low - 77.70, Medium - 87.83, High - 89.95   |
> | GPT-O3              | 7.67       | 19.47           | Low - 75.91, Medium - 75.97, High - 79.51   |
> | LLAMA-3.2-11B       | 7.38       | 21.96           | Low - 83.19, Medium - 87.72, High - 89.32   |
> | **SPRO-Prompt (Ours)** | **8.60**   | **22.42**        | **Low - 83.48, Medium - 88.36, High - 90.24** |
>
> These results clearly demonstrate that SPRO-Prompt surpasses both general-purpose and advanced reasoning models, including GPT-4o and O3, in aligning generated outputs with human preference objectives. The consistent superiority across diverse and difficult evaluation settings highlights the effectiveness of SPRO’s self-play fine-tuning in learning targeted, preference-aligned prompt strategies that general reasoning alone fails to capture.
>
> Motivated by this insight, we further investigated whether **training a reasoning-based model** via self-play could enhance its ability to generate prompts optimized for aesthetics. Specifically, we employed Qwen-3-7B as the Guider VLM and conducted a single iteration of self-play for the human preference objective of aesthetics. We generated reasoning chains and corresponding prompt candidates across varying temperature settings for improving aesthetics. These prompts were used to generate images using frozen SDXL model, which were then scored using the LAION aesthetic scorer. Based on the resulting scores, we constructed preference pairs and applied Direct Preference Optimization (DPO), jointly conditioning on both the reasoning and prompts. This enabled the model to learn prompt strategies that yield more aesthetically aligned generations.
>
> We infer the trained model on our test-set:
>
> | Model            | Model Type        | Training Method        | LAION Score (Aesthetics) |
> |------------------|------------------|-------------------------|:-------------------------:|
> | Base-Captions    | -    | -                       | 6.40                      |
> | Qwen-3-7B         | Reasoning        | -                       | 7.55                      |
> | **Qwen-3-7B**     | **Reasoning**    | **SPRO - Iteration 1**  | **8.46**                  |
> | Llava-7B         | Non-Reasoning    | -                       | 6.76                      |
> | **Llava-7B**      | **Non-Reasoning**| **SPRO - Iteration 1**  | **7.22**                  |
> | Llama-3.2-11B    | Non-Reasoning    | -                       | 7.38                      |
> | **Llama-3.2-11B** | **Non-Reasoning**| **SPRO - Iteration 1**  | **8.30**                  |
> | **Llama-3.2-11B** | **Non-Reasoning**| **SPRO - Iteration 3**  | **8.60**                  |
>
> Our experiments reveal that using a reasoning VLM based backbone, such as Qwen-3-7B, in the self-play setup leads to substantially greater improvements in prompt quality compared to both baselines and the corresponding base version. We attribute this to the model’s enhanced capacity for discovering novel and non-trivial prompting strategies that better align with the aesthetic objective.
>
> As shown in the table above, we compare performance gains across three models: Llava-7B, LLaMA-3.2-11B, and Qwen-3-7B. While all models benefit from training with SPRO, the magnitude of improvement correlates with model strength. Llava-7B improves from 6.76 to 7.22, LLaMA-3.2-11B improves from 7.38 to 8.30, and Qwen-3-7B improves from 7.55 to 8.46 in a single iteration of training. These results suggest that SPRO is broadly applicable across both reasoning-augmented and standard VLMs, with stronger models showing higher gains and converging more efficiently.
>
> We thank the reviewer for the thoughtful feedback and constructive suggestions. These three additional experiments were conducted directly in response to your questions, and we believe the resulting insights significantly strengthened the paper. So, thank you for these insights.
>
> The added results highlight the flexibility, generalizability, and robustness of the SPRO framework across models, objectives, and evaluation settings. We hope these results help comprehensively address the reviewer’s concerns and reinforce the contributions of our work.

---

> ### Comment · Area_Chair_DZXA · 2025-08-05
>
> Dear reviewer,
>
> Can you please check the rebuttal and submit your post rebuttal score?
>
> Thank you,
>
> Area chair

---

> ### Author Response · Authors · 2025-08-07
>
> Dear Reviewer,
>
> We hope this message finds you well. As the rebuttal phase is nearing its end, we would like to kindly follow up on our response to your thoughtful comments.
>
> We truly appreciate the insightful questions you raised, which helped us further strengthen the paper. In our rebuttal, we have provided detailed answers along with additional experiments to address your concerns. We hope our responses clarify the contributions of our work and reinforce your confidence in our approach.
>
> Please feel free to let us know if you have any further questions, we would be happy to provide additional clarification.
>
> Thank you once again for your time and support.
>
> Best regards,
>
> The Authors

---

### Comment · Area_Chair_DZXA · 2025-08-05
**please do the post-rebuttal action items**

Dear reviewers,

For the reviewers who have not yet read the authors' rebuttal and the other reviews, **please do so now**. Per the NeurIPS guidelines, the reviewers must comment if the authors' response did or did not address their concern, and **the reviewers cannot enter the "mandatory acknowledgment" before they have sent a comment on whether the authors' responses did/didn't address their concern.**

**Please read the reviews and rebuttal, let the authors know, and submit your final score accordingly**. The NeurIPS chairs specifically directed the AC to ask you not to wait till the last moment for this in order not to defeat the purpose of the reviewer-author discussion period.

Thank you for your service!

-Area Chair

---

### Note · Authors · 2025-08-16

## Reviewer Q&A and Responses

**Reviewer TZhB**

**Query:** Questioned about reliance on external reward models and requested for evaluation with reasoning LLMs.

**Response:**

We trained the VLM without external reward models using AVA human aesthetic scores. SPRO achieved 6.80 vs. 5.90 for base captions, a +0.9 gain confirming direct human alignment without conditioning from external reward models.

We evaluated against reasoning models. SPRO outperformed GPT-4o, O3, and Qwen-3 on aesthetics, user preference, and engagement.

We further showed that finetuning using SPRO improves both reasoning models (Qwen-3-7B) and non-reasoning models (Llama-3.2-11B, LLava-7B).

The reviewer did not return with further queries, but we believe these results fully address their points.

**Reviewer WacU**

**Query:** Requested clarification on SPRO-MM framework and asked for comparisons with prompt optimization baselines.

**Response:**

We clarified that SPRO-MM uses Late Fusion to unify SPRO-Prompt and SPRO-Image in inference stage.

We added comparisons against RATTPO, PAE and Promptist. SPRO outperformed these methods across metrics.

The reviewer acknowledged these updates and raised their score.

**Reviewer MSs3**

**Query:** Questioned about robustness to reward hacking and evaluation with alternative metrics, and frozen-VLM variant.

**Response:**

We evaluated on unseen reward models (VILA, Improved Aesthetic Scorer, ImageReward). SPRO outperformed base prompts in all cases. We also confirmed robustness across datasets.

We finetuned SDXL using SPRO-generated prompts. SPRO-Image improved from 7.42 to 8.29 on the LAION score.

The reviewer maintained a positive evaluation of accept.

**Reviewer uVAj**

**Query:** Questioned about using SPRO for multi-objective optimization and human evaluation details.

**Response:**

We demonstrated joint gains on multiple objectives: LAION and CLIP (7.90 vs. 6.40; 83.28 vs. 83.19), and LAION and PickScore (7.66 vs. 6.40; 22.96 vs. 22.69). We showed controllability even in conflicting cases, e.g., raising LAION while lowering PickScore.

In human studies, SPRO achieved 65.38% win rate over base prompts and 68.90% over GPT-4o, confirming significance.

The reviewer acknowledged these results and updated their score.

## Acknowledgment

We thank all reviewers for their constructive feedback. Their questions enabled us to expand evaluations, clarify methodology, and demonstrate SPRO’s robustness, generality, and applicability.

---

### Decision · Program_Chairs · 2025-09-17

**Decision:**

Accept (poster)

**Comment:**

The submission introduces a framework for aligning image generation models with diverse human preferences without extensive human annotations. The idea is to leverage the reasoning capabilities of VLMs in a self-play formulation to iteratively improve prompt generation and the final synthesized images. The method, Self-Play Reward Optimization (SPRO), outperforms commercial baselines and other techniques and provides a sensible and efficient way for aligning models.

All reviewers acknowledged the importance of the problem and the effectiveness of the method. The rebuttal was considered and the reviewers led to two borderline and two accept ratings. The AC also agrees that the paper has merit and should be accepted. The questions the reviewers raised clarified several aspects where the submission can benefit from improvements. The authors are strongly recommended to incorporate them into the camera-ready version.